# Structure of the human marker of self 5-transmembrane receptor CD47

Gustavo Fenalti [1✉], Nicolas Villanueva[1], Mark Griffith[2], Barbra Pagarigan[1], Sirish Kaushik Lakkaraju[3], Richard Y.-C. Huang[4], Nadia Ladygina[5], Alok Sharma[3], David Mikolon[6], Mahan Abbasian[6], Jeffrey Johnson[6], Haralambos Hadjivassiliou[6], Dan Zhu[6], Philip P. Chamberlain [2], Ho Cho[6] & Kandasamy Hariharan[6]

CD47 is the only 5-transmembrane (5-TM) spanning receptor of the immune system. Its extracellular domain (ECD) is a cell surface marker of self that binds SIRPα and inhibits macrophage phagocytosis, and cancer immuno-therapy approaches in clinical trials are focused on blocking CD47/SIRPα interaction. We present the crystal structure of full length CD47 bound to the function-blocking antibody B6H12. CD47 ECD is tethered to the TM domain via a six-residue peptide linker ([114]RVVSWF[119]) that forms an extended loop (SWF loop), with the fundamental role of inserting the side chains of W118 and F119 into the core of CD47 extracellular loop region (ECLR). Using hydrogen-deuterium exchange and molecular dynamics simulations we show that CD47's ECLR architecture, comprised of two extra-cellular loops and the SWF loop, creates a molecular environment stabilizing the ECD for presentation on the cell surface. These findings provide insights into CD47 immune recognition, signaling and therapeutic intervention.

[1] Molecular Structure and Design, Bristol Myers Squibb, San Diego, CA, USA. [2] Protein Homeostasis, Bristol Myers Squibb, San Diego, CA, USA. [3] Molecular Structure and Design, Bristol Myers Squibb, Princeton, NJ, USA. [4] Pharmaceutical Candidate Optimization, Nonclinical Research and Development, Bristol Myers Squibb, Princeton, NJ, USA. [5] Pharmacology, Bristol Myers Squibb, San Diego, CA, USA. [6] Discovery Biotherapeutics, Bristol Myers Squibb, San Diego, CA, USA. ✉email: Gustavo.Fenalti@bms.com

Cluster of Differentiation 47 (CD47) is an integral membrane receptor containing a heavily glycosylated N-terminal IgV-like extracellular domain (ECD), a 5-transmembrane (5-TM) spanning helical bundle domain, and a small, C-terminal domain (CTD). The CTD is alternatively spliced and found as four isoforms, ranging from 4 to 36 residues[1–3]. CD47 is ubiquitously expressed in human cells, including red blood cells (RBCs) and platelets, and the expression of its different isoforms is tissue specific[4]. CD47 is an essential component of the innate immune system, and binding of the ECD to signal regulatory protein alpha (SIRPα)[5–7], abundant in myeloid cells and particularly on macrophages, activates a signaling response that inhibits cell phagocytosis ('don't eat me' signal). Tumor cells hijack this signaling mechanism and overexpress CD47 to evade the immune system and enhance survival[6]. A promising approach in cancer immuno-therapy consists of blocking CD47-SIRPα interaction, and its therapeutic potential (as a monotherapy or in combination) is currently under investigation in multiple clinical trials.

Pioneering studies identified B6H12, a function-blocking anti-CD47 monoclonal antibody (mAb) that can inhibit the binding of two endogenous partners, SIRPα and the secreted glycoprotein Thrombospondin-1 (Tsp-1); B6H12 showed efficacy in pre-clinical tumor models and hematological malignancies[2,8–13]. Although SIRPα-CD47 represents a major immune checkpoint signaling axis, the biology of CD47 is complex and multifaceted. In addition to SIRPα, the extracellular glycoprotein Tsp-1, which belongs to a family of proteins with antiangiogenic functions, and a number of integrins like the heterodimeric αVβ3, also bind to CD47[2,14,15]. When bound to Tsp-1 or integrins, CD47 was found to form larger complexes and co-precipitate with heterotrimeric G proteins, but the details of these intracellular protein associations remain largely unknown[16,17]. Other intracellular CD47 protein partners have been described, most notably the association with protein 4.2 within the band 3 macro-complex in RBC[4,18]. Altogether, association of the distinct CD47 domains to a myriad of endogenous protein ligands activates different signaling pathways controlling cardiovascular homeostasis, cell proliferation and differentiation, angiogenesis and immune regulation[17,19,20].

Several families of integral transmembrane receptors like the 4-TM tetraspanins, G protein-coupled receptors (GPCRs) and CD20-like receptors, are comprised of homologous members that evolved from a common ancestor to exert specific functions in human physiology. In contrast, CD47 is the only known 5-TM receptor member of the immune system, and has a molecular machinery adapted to exert multiple roles in signal transduction across cell membranes. Together with the distantly related members of the prominin family, CD47 is the only 5-TM receptor of the human genome, and no structural information is available for the transmembrane domain of these receptors. Like other receptors of the immune system, the cd47 gene is only present among higher vertebrates, and the amino acid conservation of the SIRPα/CD47 binding interface is species specific, as evidenced by the different levels of cross reactivity between species (e.g., human SIRPα can bind CD47 from human and pig sources, but does not bind mouse or rat)[21]. Interestingly, members of the Poxviridea family of viruses, which devote numerous genes to the expression of molecules for evasion of the host immune system, express CD47-like receptors that have amino acid similarity to CD47 receptors from some vertebrate species[22].

Given the emergence of immuno-oncology therapeutics that target CD47, and its numerous biological roles in health and disease, a structural characterization of the full- length receptor is needed. To better understand the atomic features associated with CD47 immune recognition and transmembrane signaling[9,13,23],

we determined the crystal structure of the full-length human CD47 in complex with the Fragment antigen binding (Fab) of the mAb B6H12. This structure provides atomic details of a unique 5-TM receptor fold and reveals key interactions in the extracellular loop region (ECLR) that maintain CD47 ECD orientation on the surface of cells. Mutagenesis and kinetic hydrogen-deuterium exchange mass spectrometry (HDX-MS) data revealed that residues in the extracellular loop (ECL) 1 and 2 have an important role stabilizing the inter-domain peptide linker [114]RVVSWF[119], connecting the ECD to the TMD. Further, our computationally determined mechanism suggests the ECD mobility is facilitated by the 'hinge' peptide sequence [114]RVVSWF[119], and the position of a key 'conformational switch' residue Y184. These data provide insights into CD47 ECD 'self' recognition, transmembrane signaling and cancer therapy.

## Results

**Overall architecture of the CD47[BRIL]-B6H12 complex**. To facilitate crystallization of CD47 in lipidic mesophases[24] we engineered a construct consisting of the full-length human CD47 (residues 1–305, isoform 1) with a thermostabilized (M7W, H102I, and R106L) apocytochrome b562RIL from *Escherichia coli* (BRIL)[25] fusion protein inserted in the intracellular loop (ICL) 1 of the receptor, namely CD47[BRIL] (Methods). We crystallized and determined the 3.4 Å resolution crystal structure of CD47[BRIL] in complex with the Fab from the mAb B6H12 (CD47[BRIL]-B6H12) (Fig. 1a and Supplementary Table 1; Methods). The crystal-lographic asymmetric unit contains a dimer of the CD47[BRIL]-B6H12 assembly, and interactions between the two CD47[BRIL]-B6H12 units are mediated entirely through the Fab's from each unit (Supplementary Fig. 1b). While most of the CD47 residues from the ECD and TMD were modeled for one chain of the receptor (residues 1–278), residues 211–215 and 279–305, equivalent to ICL2 and CTD respectively, were completely disordered and not visible in the electron density maps. The other chain of the receptor displayed interpretable electron density for the ECD only (residues 1–117), and residues 118–305 could not be unambiguously resolved in the maps and hence were not modeled. The BRIL fusion proteins were completely disordered and not visible in the electron density maps. The overall structure of the CD47[BRIL]-B6H12 complex exhibits an elongated conformation, with the Fab B6H12 recognizing a monomeric CD47 receptor with an epitope on the most apical region of the ECD (Fig. 1a). The crystal structure of the N-terminal soluble CD47 ECD (residues 1–115) in complex with the B6H12 Fab[26], PDB 5TZU [https://doi.org/10.2210/pdb5TZU/pdb], was previously determined and is nearly identical to the equivalent atoms of the CD47[BRIL]-B6H12 structure presented here (r.m.s.d. of 1.4 Å over 540 Cα atoms) (Supplementary Fig. 1c, d). The CD47 ECD comprises the known IgV-like fold, and a six-stranded mainly parallel AGFCC'C" β-sheet harboring the binding site of SIRPα, oriented at an ~80° angle with respect to the TMD and to the membrane environment (Fig. 1a, Supplementary Fig. 1 and Methods). Electron density demonstrating N-linked glycosylation was identified for all five CD47 glycosylation sites within the ECD (N5, N16, N32, N55, and N93), but only modeled for N16, N32, N55, and N93; N16 is positioned close to the membrane environment in the vicinity of the ECL2 and the inter-domain C15-C245 disulfide bond (Fig. 1b).

**The [117]SWF[119] loop links the ECD to the ECLR**. Despite the progress made towards understanding CD47 signaling mechanisms, the atomic details of the interactions between CD47 ECD and the rest of the receptor remained elusive. The CD47[BRIL]-B6H12 complex structure reveals the presence of a six-residue

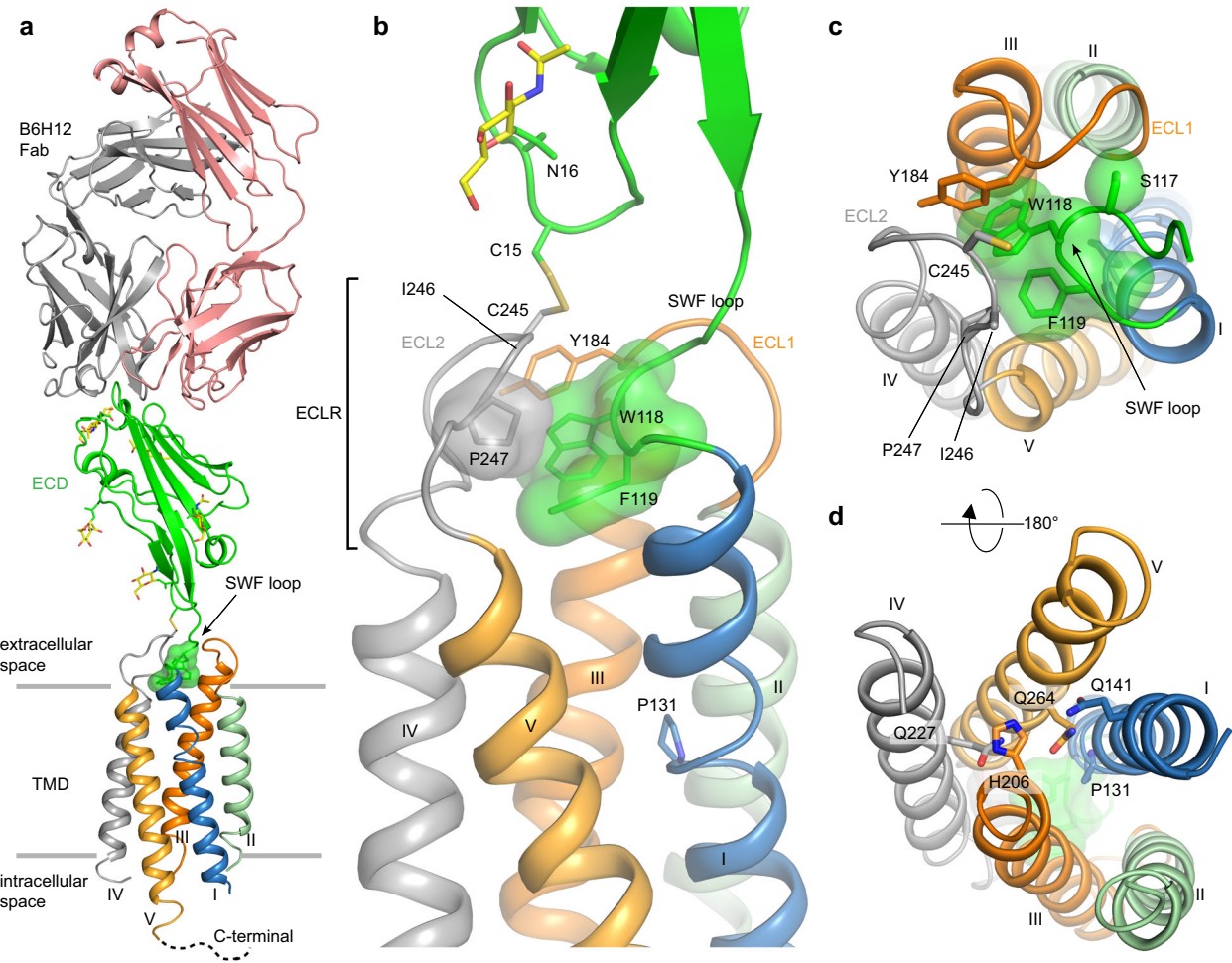

**Fig. 1 Overall structure of CD47<sup>BRIL</sup>-B6H12 complex, ECLR, and TMD. a** CD47<sup>BRIL</sup>-B6H12 complex structure showing the Fab (light and heavy chains shown as pink and gray cartoon respectively), the ECD (green cartoon), and the TMD helices (helix I, blue; helix II, light green; helix III, orange; helix IV, gray and helix V light orange). The CD47 ECLR is formed by the ECL1 (orange tube), ECL2 (gray tube), and the ¹¹⁷SWF¹¹⁹ loop (green surface shown for W118 and F119). The N-linked glycans are shown as sticks with yellow carbons. The gray lines represent the approximate extracellular and intracellular membrane boundaries. **b** Close up view of the receptor ECLR showing all the three ECLR loops and important residues (side chains shown as sticks and transparent surfaces). The location of the helical kink in helix I centered on P131 (side chain shown as sticks) is also shown. **c** Top view of CD47 TMD bundle from the extracellular side showing the position of helices and orientation of the three extracellular loops with respect to the helices. **d** View of the TM bundle from the intracellular side. The side chain of residues in the IC hydrogen bond network are shown as sticks and labeled.

linker between the C-terminal end of the ECD and the N-terminal extracellular tip of helix I. This linker is comprised by the conserved sequence motif ¹¹⁴RVVSWF¹¹⁹, and forms an extended loop referred hereafter as the ¹¹⁷SWF¹¹⁹ loop (Figs. 1a, b, c, 2a and 3a, b). The ECLR of CD47 is therefore composed of three loops: the ¹¹⁷SWF¹¹⁹ loop, the ECL1, and the ECL2 which harbors C245, a key residue required for a conserved functional inter-domain disulfide bond with ECD C15²⁷ (Figs. 1b, c, 2b and 3a). The three ECLR loops, through intricate non-covalent interactions among them, together with the inter-domain C15-C245 disulfide bond, provide the molecular environment that supports the ECD orientation (Figs. 1a, b, 2a, b, and 3a, b). The hydrophobic aromatic side chains of W118 and F119 on the ¹¹⁷SWF¹¹⁹ loop are completely buried (319Å²) deep in the center of the ECLR, and form a tilted (45°) edge-to-face aromatic interaction between them (Figs. 1b, c, 2a, and 3a). Notably, the ¹¹⁷SWF¹¹⁹ loop is nearby ECL2, and W118 and F119 side chains are positioned immediately below the ECL2 residues C245, I246, and P247, within van der Waals contact distance (Fig. 1b, c). This molecular arrangement provides a stable environment for the

ELC2, and consequently for the inter-domain disulfide bond between C245, at the center of the loop, and C15 in the ECD (Fig. 1b). The insertion of the ¹¹⁷SWF¹¹⁹ loop residues into the center of the ECLR, together with the C15-C245 inter-domain disulfide bond are the two 'anchor' points tethering the CD47 ECD to the ECLR. Our structural data are consistent with previous CD47 mutagenesis studies where substitution of C245 or C15 to serine, preventing disulfide bond formation, significantly reduced SIRPα binding, cell–cell adhesion, and signal transduction across cell membranes as measured by Ca⁺² signaling studies²⁷.

**Y184 position bridges the ECLs and ¹¹⁷SWF¹¹⁹ loop.** A unique structural feature of CD47 ECLR is its molecular architecture and the intimate interactions between all three loops (ECL1, ECL2, and the ¹¹⁷SWF¹¹⁹ loop). In the CD47<sup>BRIL</sup>-B6H12 structure, the extracellular portion of helices II and III, flanking the ECL1, are juxtaposed remarkably close to one another (Figs. 1c, 2b and 3c, d, e). This close proximity is facilitated by two conserved glycine residues (G175 and G191) that meet at the interface between helices II and

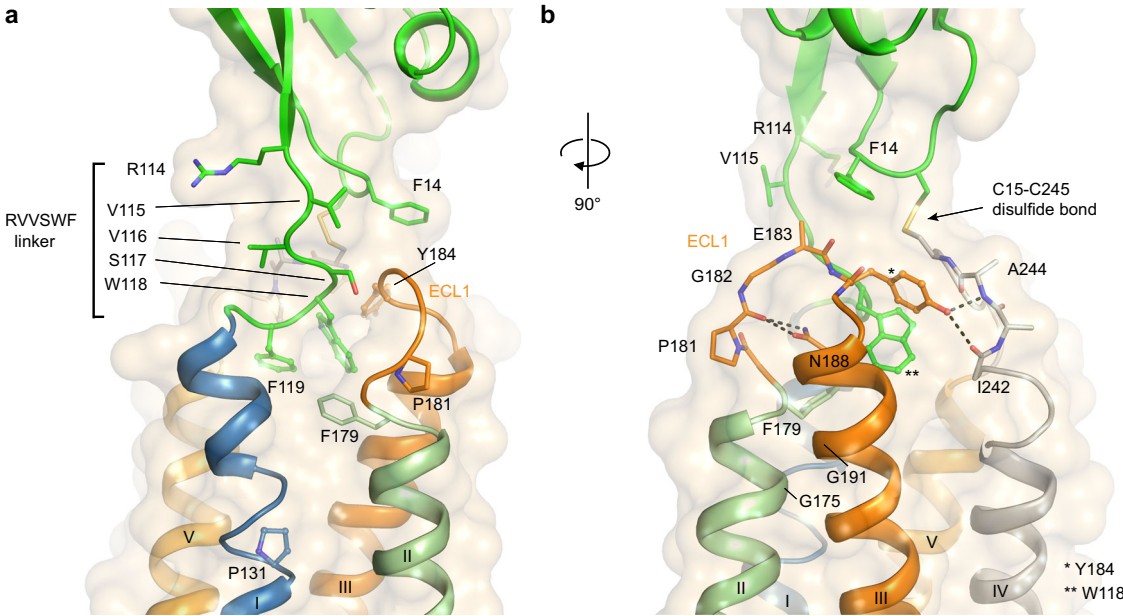

**Fig. 2 Interactions within the ECLR and $^{114}$RVVSWF$^{119}$ linker. a** View of the $^{114}$RVVSWF$^{119}$ six-residue peptide linker in the CD47$^{BRIL}$-B6H12 crystal structure (green cartoon and sticks), and the insertion site of the $^{117}$SWF$^{119}$ loop in the ECLR core. The ECD is shown as green cartoon and the TMD helices are colored blue (helix I), light green (helix II), orange (helix III) and light orange (helix V). The ECL1 is shown as orange cartoon and sticks. **b** A 90 degree rotation of CD47$^{BRIL}$-B6H12 structure view shown in (**a**) highlighting the ECL1 hydrogen bonds (black dotted lines) between Y184 side chain and ECL2 (main-chain atoms shown as gray sticks and labeled), and between N188 side chain and ECL1 (orange sticks).

III; this arrangement of glycine residues between helices is a feature also present in the structure of the CD20 receptor[28], a member of the four transmembrane (4-TM) family of immune receptors (Fig. 2b). The short CD47 ECL1 is comprised of six residues (V180-S185) with a flexible glycine residue (G182) located in the middle of the loop, allowing for additional flexibility at the tip of ECL1, particularly for the adjacent residues E183 and Y184 (Fig. 2b). Remarkably, the surface exposed Y184 side chain rests in a large pocket at the top of the CD47 ECLR that is formed in between the ECL1 and ECL2, and has the $^{117}$SWF$^{119}$ loop residue W118 at the bottom (Figs. 1b, c, 2b, 3a and Supplementary Fig. 2). Further, the ECL1 loop conformation and positioning of the Y184 side chain precisely between the ECL1 and ECL2 represents the only contact point connecting ECL1 and ECL2 (Figs. 1b, c, 2b and 3a). Accordingly, the Y184 side chain hydroxyl moiety provides critical polar contacts bridging these loops and is at hydrogen bond distance to the main-chain carbonyl of the ECL2 I242 (2.9 Å), and to the main-chain amide of A244 (3.0 Å) (Figs. 2b and 3a). In addition to these interactions, Y184 side-chain atoms pack directly against the disulfide forming C245 residue, thus contributing to the stabilization and proper orientation of the inter-domain C15-C245 disulfide bond of CD47 (Figs. 1b, c, 2b, and 3a).

**CD47 helix III is at the center of the 5-TM bundle.** The TMD has a fold composed of five TM helices forming a helical bundle that spans ~32 Å across the lipid membrane and adopts a 'pentagon-like' shape when viewed from the outer leaflet of the membrane, with each helix at a vertex (Fig. 1c). Each helix of the TMD adopts a distinct angle relative to the membrane environment. Helix II is nearly perpendicular to the lipidic membrane environment, while helices I, III, IV, and V span the membrane at various shallower angles (Fig. 1a, Supplementary Fig. 3a). The extracellular side of the TM helices are further apart from each other compared to the intracellular (IC) side. This is mostly due to a proline induced kink at P131, in the middle of TM1, allowing for a helical tilt away from

the receptor core, and the angle of helix III, positioned in the middle of the TM bundle at the IC side (Figs. 1b, c, d and 2a).

Overall, the CD47 TMD in the CD47$^{BRIL}$-B6H12 crystal structure exhibits an extensive hydrophobic TM core, with a total of 22 side chains from hydrophobic residues along the length of helices I–V, pointing into the center of the receptor (Fig. 3). Notably, a cluster of hydrophobic residues (hydrophobic cluster 1, HC1) occupy the entire extracellular portion of the TMD core, completely occluding the center of the receptor TMD (Fig. 3). The HC1 spans a region from the outer leaflet of the TMD, to the center of TMD and lipid bilayer, a position marked by the presence of the polar residue T199 (Fig. 3a, b). The 5-TM bundle of CD47 adopts a fold where the most extracellular helical turns of each TM helix places a hydrophobic side chain into the center of the helical bundle, forming a layer of tightly packed hydrophobic side chains (helix I: I127; helix II, F179; helix III L192; helix IV: L238; helix V: L253) (Fig. 3c). These five residues form the contiguous surface that creates the interface between the TMD and the ECLR (Fig. 3a, b and c). A similar layered arrangement is observed for the remaining hydrophobic residues in the HC1, forming a second and third layer of hydrophobic packing, although the packing in the receptor core appears to loosen further from the ECLR (Fig. 3c, d and e). A second smaller cluster of hydrophobic residues (hydrophobic cluster 2, HC2) is found at the intracellular side and is restricted to a region between helices I, II, and III (Fig. 3a).

Helix III in the CD47$^{BRIL}$-B6H12 crystal structure adopts a central position in the TM bundle and displays several features that highlight its key structural importance within the CD47 TMD bundle. It has a unique transversal orientation, assuming a more distal position in the EC side, and is inserted into the center of the TM bundle at the IC side (Fig. 1c, d, and Supplementary Fig. 3). Helix III is the only helix of the bundle that forms inter-helical hydrophobic contacts with each of the other four TM helices. It buries a total of 1,373Å$^2$ of its hydrophobic surface in the receptor TMD core, with the side chain of residues L192,

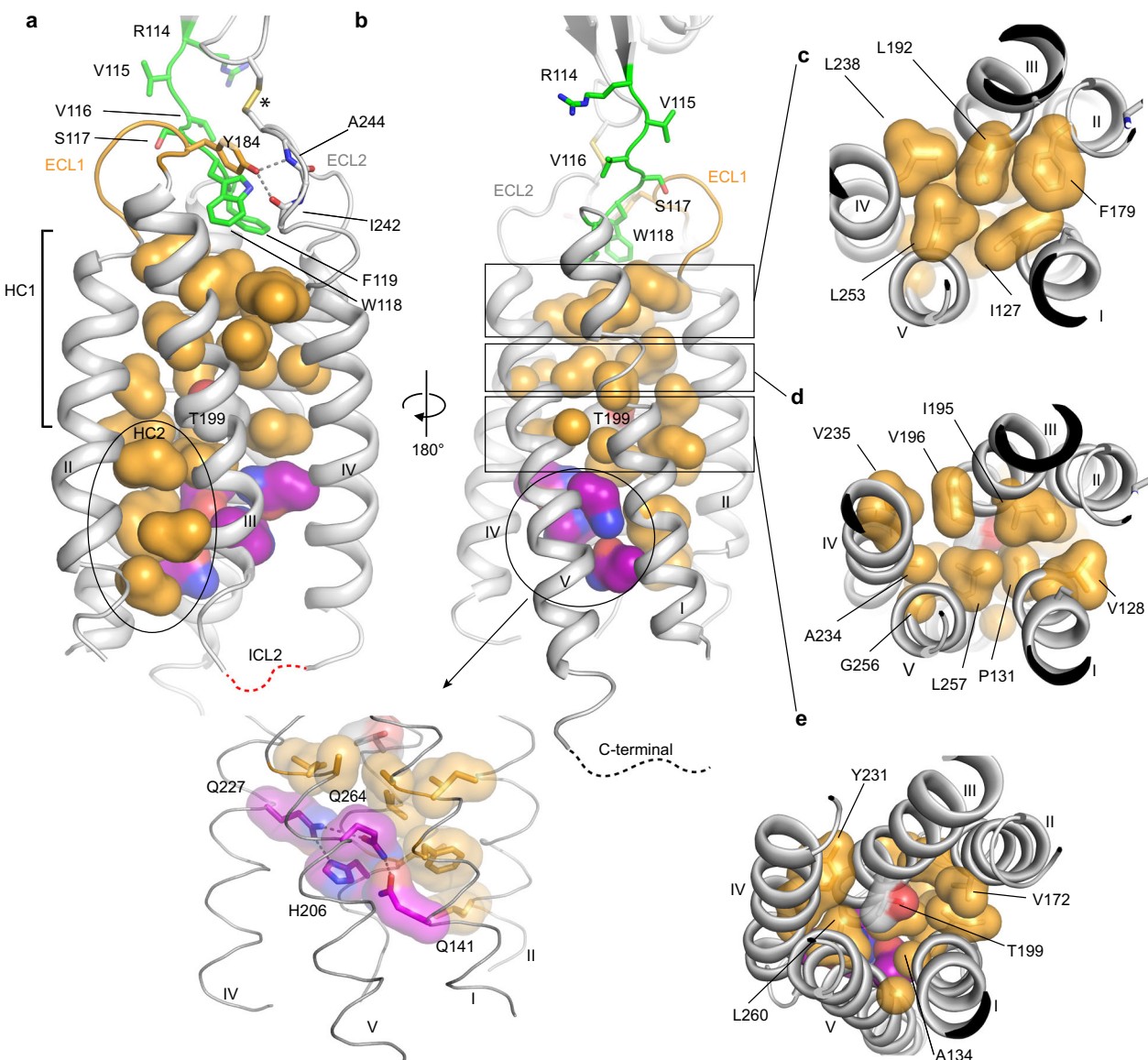

**Fig. 3 Physicochemical characteristics of CD47 TMD core. a–b** Two views of the TMD of CD47^BRIL-B6H12 crystal structure (gray cartoon, helices numbered I-V, ECD and Fab atoms omitted) showing residues in the core of the TMD helix bundle and in the ECLR. The surface of TMD hydrophobic residues pointing towards the core of the receptor are shown in light orange. The side chains of residues in the ^114RVVSWF^119 linker are shown as green sticks. Hydrogen bonds are represented by gray dotted lines. The asterisk indicates the position of the inter-domain disulfide bond between C15 and C245. Black ellipses delineate the IC region containing the cluster of hydrophobic residues between helices I, II, and III (HC2, shown in (**a**)), and residues involved in the IC hydrogen bond network (shown as purple transparent surfaces and sticks in (**b**)). ECL1 and ECL2 are colored orange and gray respectively and are labeled. Disordered ICL2 and C-terminal residues are represented by red and black dotted lines respectively. **c–e** Top views from the CD47 EC side showing the packing of hydrophobic residues in the receptor core and the different layers (delineated by the rectangles in (**b**)), from the outer membrane leaflet to the center of lipid bilayer.

I195, S198, T199, and L202 displaying the highest buried surface areas among all residues in the receptor TMD (Supplementary Fig. 3). A notable break in the otherwise entirely hydrophobic and packed TMD (with the exception of Ser198 and T199), is the presence of five polar residues from helix I (Q141, K145), helix III (H206), helix IV (Q227), and helix V (Q264) near the IC side of CD47 TMD, in the region adjacent to HC2 (Fig. 3a, b). We observe a hydrogen bond network centered on helix III H206, and involving Q141, Q227, and Q264 (Fig. 3b). The hydrogen bond interactions connecting helices I, III, IV, and V may contribute to the local stability of the CD47 TMD bundle at the IC side of the membrane, closer to the ICL's, and the CTD where intracellular protein partners are reported to bind[15,29].

**Evolutionary conservation of the 5-TM receptor CD47.** Sequence and structural database searches using the TMD of CD47^BRIL-B6H12 crystal structure revealed no substantial sequence or structural similarities to any other protein, thus suggesting that CD47 adopts a unique 5-TM fold (Methods). CD47 evolutionary analysis data was therefore leveraged to aid in the identification of conserved amino acid regions of the receptor across different species, and to highlight structural motifs likely associated with receptor function or required for structural integrity. We analyzed a non-redundant set of CD47 amino acid sequences from all available CD47 expressing species, including those from viruses of the *Poxviridae* family that encode a CD47-like receptor (Supplementary Figs. 4, 5, and 6). Overall, the

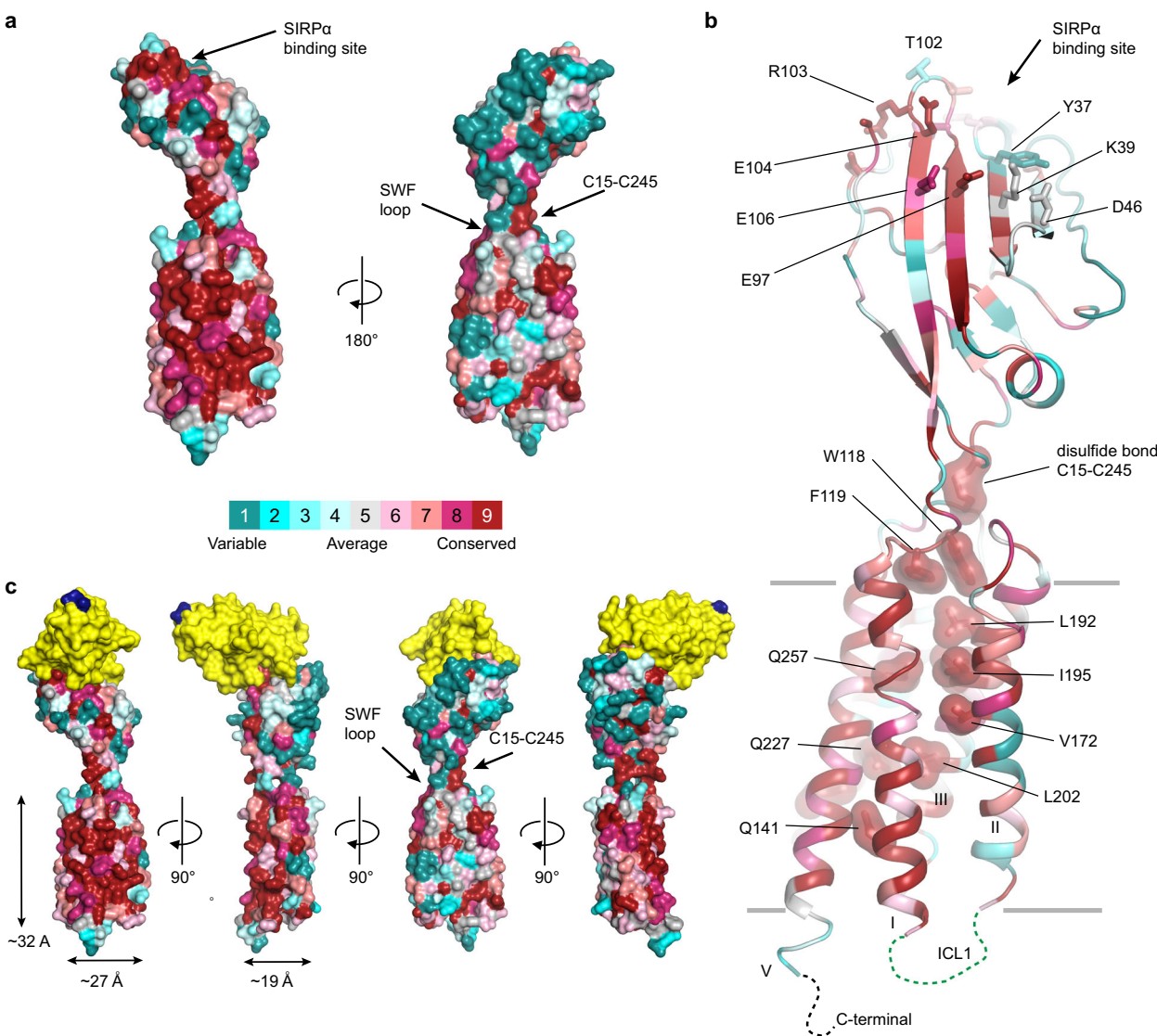

**Fig. 4 Amino acid evolutionary conservation of CD47 and the SIRPα binding site. a** Two views of the CD47$^{BRIL}$-B6H12 crystal structure (Fab atoms omitted) with the mammalian CD47 amino acid conservation (Supplementary Fig. 4; source data provided as a Source Data file) mapped on the surface representation of the receptor. The color scheme for the conservation scores is shown as a colored bar. **b** Cartoon representation of the CD47$^{BRIL}$-B6H12 crystal structure (Fab atoms omitted) colored according to the conservation scale bar presented in (**a**). The side chains of highly conserved residues with a score of 9 in the conservation scale, located in the core of the TMD, in the ECLR or IC hydrogen bond network are shown as dark red surfaces. The side chains of C15 and C245, are also shown as dark red surface. CD47 residues that form the SIRPα binding site are shown as sticks; some non-conserved residues in the vicinity of the epitope are also shown. Gray lines represent the approximate lipid membrane boundaries. Disordered ICL1 and C-terminal residues are represented by green and black dotted lines respectively. Amino acid evolutionary conservation data are provided in Supplementary Fig. 4. Source data are provided as a Source Data file. **c** Superposition of the full-length CD47$^{BRIL}$-B6H12 crystal structure (Fab atoms omitted) with the crystal structure of the soluble CD47 ECD in complex with SIRPα, PDB ID 2JJT [https://doi.org/10.2210/pdb2JJT/pdb]. The ECD of both CD47 structures were used for the structural alignment. The full-length CD47 is shown as a surface representation and colored as in (**a**); the SIRPα is shown as yellow surface. The last C-terminal residue of SIRPα domain 1 is colored blue for reference.

pairwise sequence alignment and phylogenetic analysis revealed that CD47 from mammals retained substantially higher amino acid identity, with 62 of 305 residues absolutely conserved among humans and all other 71 mammalian species (Supplementary Figs. 4 and 5, and Supplementary Data 1). These include residues W118 and F119 in the $^{117}$SWF$^{119}$ loop, residues contributing to the packing of the hydrophobic TM bundle core, and residues in contact with residues in the ECLR (Fig. 4b, Supplementary Fig. 4 and Supplementary Data 1). Importantly, mapping the mammalian amino acid sequence conservation onto the full-length

CD47 crystal structure revealed a highly conserved contiguous surface formed by the ECD and TMD, which spans the SIRPα binding face of the ECD, through the membrane-spanning portion, primarily centered on helix I and helix V (Fig. 4a and c, Supplementary Fig. 4 and Methods). In contrast, a 180° rotation of the receptor (on the opposite face of the ECD binding site for SIRPα) reveals a largely non-conserved surface from the ECD, through helices II, III and IV in the TMD (Fig. 4a and c, and Supplementary Fig. 4). As expected, the core of the IgV-like ECD domain also displays higher conservation, however, the SIRPα

binding site is only partially conserved, which is in agreement with the knowledge of the different CD47-SIRPα cross-species reactivity[21] (Fig. 4b, Supplementary Fig. 4).

Sequence conservation is lower when examining the more divergent clade in the CD47 phylogenetic tree comprised of 59 avians and reptiles, with only 17 out of 305 residues being identical to the human sequence (Supplementary Fig. 5 and Supplementary Data 2). The most divergent branch in the phylogenetic tree is formed by CD47-like receptors from viruses (22 sequences), which display the lowest overall amino acid identity compared to human CD47. Only seven identical residues are found, including the inter-domain disulfide bond pair C15-C245, and a highly conserved glycine (G191, human numbering) located on the top extracellular portion of helix III; in mammals, and vertebrates in general, this residue is part of a highly conserved $^{191}$GLG$^{193}$ sequence motif that packs against helix II residues and inserts L192 in the middle of the 5-TM core, precisely bellow the $^{117}$SWF$^{119}$ loop (Supplementary Fig. 6, Fig. 2b, Fig. 4 and Supplementary Data 3). Further, a direct analysis of the human CD47 amino acid sequence and the closely related viral orthologues (12 sequences) revealed 55 identical residues. They are clustered in the extracellular portion of the 5-TM bundle and ECLR, and more scattered in the ECD. Despite being very distantly related in the evolutionary scale, it is remarkable that human and CD47-like receptors from viruses share amino acid identity in key regions of the receptor, such as residues 118W and 119F in the $^{117}$SWF$^{119}$ loop, residues in the HC1 (F179, L192, L253, and L257) that interface the ECLR, and the inter-domain disulfide bond (Supplementary Fig. 6 and Supplementary Data 4). Collectively, these data support the findings that residues in these regions are crucial for the fold and function of the 5-TM bundle of CD47.

**ECL residues controlling $^{114}$RVVSWF$^{119}$ linker dynamics**. To complement insights gained through analysis of the static crystal structure of CD47$^{BRIL}$-B6H12 with dynamic information, we performed differential kinetic HDX-MS studies using wild type (WT) CD47$^{BRIL}$ and CD47$^{BRIL}$ constructs containing single point mutations (Fig. 5, Supplementary Fig. 7, Supplementary Table 2 and Supplementary Data 5). Overall, the HDX profile of the WT CD47$^{BRIL}$ revealed that the C-terminal part of the ECL1 (residues 184–185), and almost the entire helix III (residues 186–202), displayed very low HDX levels (peptides 184–194 and 195-202), indicating this region adopts a highly stable conformation (Fig. 5). The restricted conformational dynamics of this region is consistent with its position in the center of the receptor TMD bundle, where several side chains from helix III (L192, I195, T199, V196, and L202) are buried in the receptor core, while N188 makes hydrogen bond interactions with the ECL1 residue P181 (Fig. 2b and Supplementary Fig. 3). In contrast, the region spanning the inter-domain $^{114}$RVVSWF$^{119}$ linker and the first two EC helical turns of helix I (peptides 111–126, 112–126, and 114–126) showed much higher HDX levels, illustrative of the higher conformational dynamics of this region (Fig. 5, Supplementary Fig. 7 and Supplementary Data 5). Despite harboring the inter-domain disulfide bond between C15-C245, the HDX levels of the ECL2 and EC portion of helix V (peptide 244–253) slightly exceeded the levels observed for the inter-domain $^{114}$RVVSWF$^{119}$ linker (peptides 111–126, 112–126 and 114–126) containing the $^{117}$SWF$^{119}$ loop residues partially buried in the ECLR core (Fig. 5, Supplementary Fig. 7 and Supplementary Data 5).

Mutation of the ECL2 residue C245 into a serine prevents the formation of the inter-domain disulfide bond (C15-C245), impairs receptor function[27], and resulted in an increase in the HDX exchange, indicative of an increase in the conformational

dynamics of the inter-domain $^{114}$RVVSWF$^{119}$ linker (peptides 111–126, 112–126, 114–126 and 119–126), with negligible HDX changes on other characterized regions of the receptor (Fig. 5d, Supplementary Fig. 7 and Supplementary Data 5). Notably, mutation of Y184 in the ECL1 into a phenylalanine, thus removing the hydrogen bond potential of Y184 side chain to the backbone atoms of ECL2 residues A242 and A244, likewise resulted in an increase of HDX levels in the inter-domain $^{114}$RVVSWF$^{119}$ linker (peptides 111–126, 112–126, 114–126 and 119–126), although to a lesser extent compared to C245. In addition, removal of a phenyl aromatic side chain at position 184 (Y184A mutant) leads to a further increase in HDX uptake and destabilization of the inter-domain $^{114}$RVVSWF$^{119}$ linker compared to the Y184F mutant (peptides 111–126, 112–126, 114–126 and 119–126) (Fig. 5d, Supplementary Fig. 7 and Supplementary Data 5). This result reflects loss of the side chain interaction between Y184 and W118 in the ECLR core, as well as the backbone hydrogen bond interactions with ECL2 residues (A242 and A244) (Fig. 5, Supplementary Fig. 7 and Supplementary Data 5).

Although substitutions of W118 or F119 by an alanine hampered the pairwise comparison of HDX kinetics with other constructs in the $^{117}$SWF$^{119}$ loop region due to mutation-induced changes in digestion profile and HDX rate, W118A resulted in the highest HDX increase in the EC portion of helix I among all mutants (peptide 119–126, 42.5% at 60 min) (Supplementary Fig. 7 and Supplementary Data 5). This finding highlights an important role for the $^{117}$SWF$^{119}$ loop residue W119, stabilizing the EC portion of helix I, a region immediately adjacent to the $^{117}$SWF$^{119}$ loop, which is tilted away from the TMD bundle in the CD47$^{BRIL}$-B6H12 structure. The HDX profile for the mutant F119A lacked common peptides covering $^{117}$SWF$^{119}$ loop region and EC portion of helix I with respect to other mutants, therefore no differential HDX analysis could be made for this mutant.

**CD47 macrostates**. To gain insights into the conformational flexibility of the human CD47, we performed 1μs molecular dynamic simulations using the CD47$^{BRIL}$-B6H12 structure with the Fab B6H12 atoms removed. The overall fold of the IgV-like ECD domain and the TM helix bundle remained mostly unchanged during the simulations (r.m.s.d. of 1.1 Å and 1.2 Å for the ECD and TM helices Cα atoms respectively). However, large rigid body shifts of the ECD relative to the TMD were observed early on during the simulations, with stable conformations of this domain in the latter stages (Fig. 6). An analysis of the ECD trajectory in all simulations revealed a hinge-type motion where the ECD specifically 'tilts' toward the membrane, in the direction of helices I and V, and away from helices II and III (Fig. 6a, c and d). The ECD rigid body motion is unilateral and occurs primarily in one plane, and up to ~35 Å shifts in the position of the AGFCC'C" β-sheet containing the SIRPα binding site are observed (Figs. 4b, c, 6a, and Methods). Analysis of the ECD orientations during the molecular dynamics simulations shows that this domain adopts two low energy macrostates, s1 and s2 (Figs. 6b, c, d, e, and 7a). The first state (s1) is characterized by a broader peak on the ECD positioning distribution plot, with an average domain angle of ~17° from the initial 80° position observed in the crystal structure, therefore positioning the ECD at ~62° angle with respect to the TMD and membrane environment (Figs. 6b, e, 7a, b, e and g). The second state (s2) has a higher peak with an average domain orientation of ~40° from the crystal structure position, resulting in a final ~40° tilt relative to the TMD (Figs. 6b, e, 7a, c, f and h).

The transition from the s1 to s2 conformational state is associated with a subtle backwards movement of the short ECL1

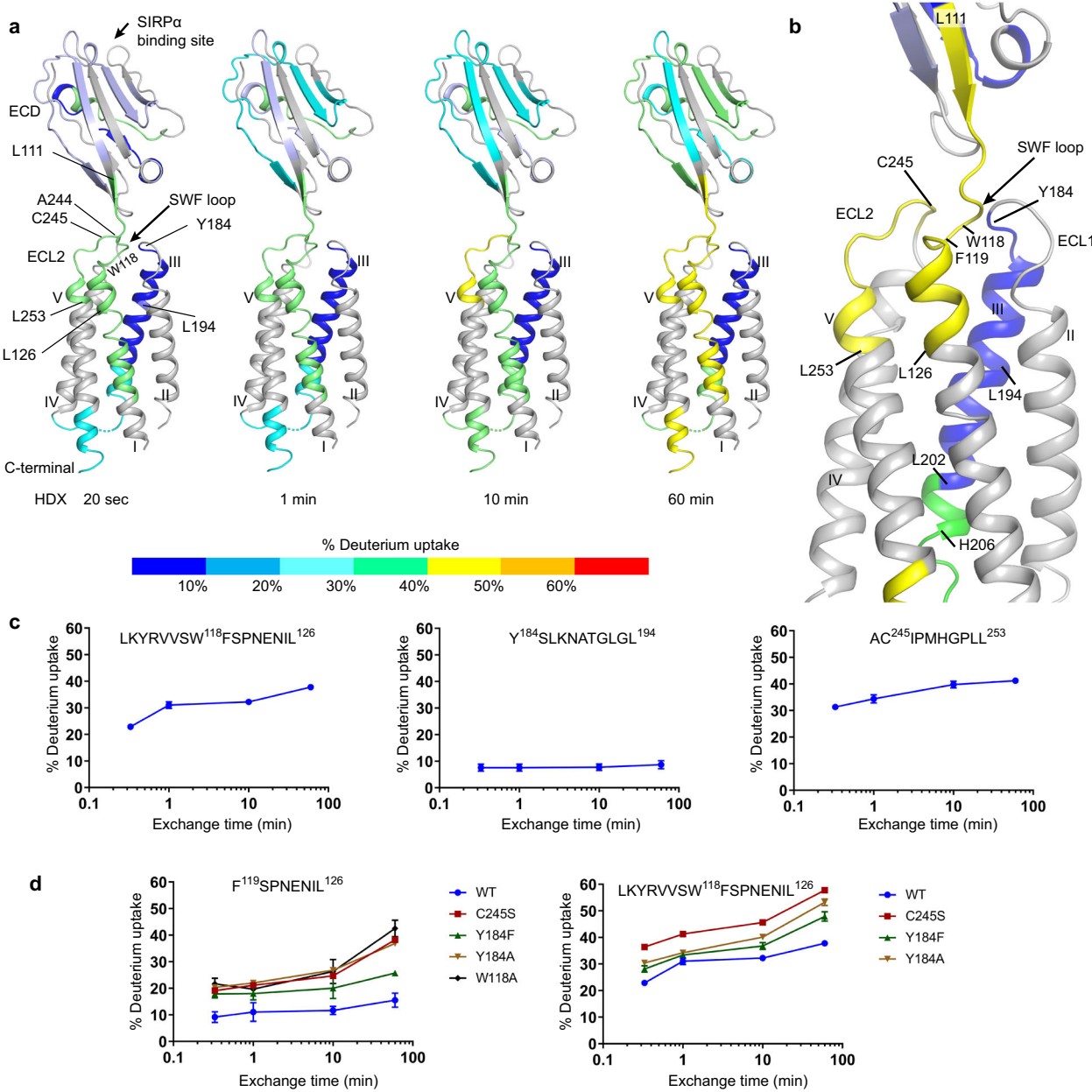

**Fig. 5 Kinetic HDX-MS data of WT CD47^BRIL and mutants. a** The HDX levels of WT CD47^BRIL and mutants are color coded and mapped onto the crystal structure of full-length CD47^BRIL-B6H12 complex (Fab B6H12 atoms omitted). Regions that were not identified in the HDX-MS experiments are colored gray. **b** Close view of the ECLR showing deuterium uptake at 60 min. **c** Deuterium uptake plots of peptides corresponding to the inter-domain ^114RVSSWF^119 linker region, ECL1 and helix III, ECL2 and the EC portion of helix V. **d** Deuterium uptake plots of WT CD47^BRIL and mutants for peptides covering the ^114RVSSWF^119 linker region. Each data-point is presented as mean values across two replicates. Error bars represent mean values ± SD of duplicate measurements. At least two independent experiments were performed with a panel a receptor constructs. The kinetic differential HDX data are provided in Supplementary Data 5.

away from the center of the ECLR, facilitated by a flexible glycine residue (G182) at the tip of the loop, and subsequent formation of a salt bridge interaction between the adjacent ECL1 E183 and K187 in helix III (Supplementary Figs. 8 and 9). The salt bridge formation occurs early during the simulations (0.2μs), and importantly, is a precursor for larger scale movements consisting of (1) egress of the Y184 side chain from the large ECLR pocket, and (2) concomitant ECD rigid body shifts and transition to the s2 state (Figs. 6, 7 and Supplementary Figs. 8 and 9). Tracking the atom distances between the Y184 side chain hydroxyl group and the main-chain atoms of I242 and A244 (hydrogen bonded in the crystal structure, Y184 'in' position) revealed that Y184 side-chain

conformation switches to an alternative Y184 'out' conformation at ~0.3 μs (Fig. 7b, Supplementary Figs. 8 and 9). The Y184 position switch is accompanied by the ECD rigid body shifts and transition of this domain to the s2 state at ~0.3 μs (Fig. 7b, c, and Supplementary Figs. 8 and 9). Collectively, our data suggest that departure of Y184 from the ECLR pocket, where it hydrogen bonds to the ECL2 and packs against the C15-C245 disulfide bond, is associated with a conformational change that results in a ~22° tilt of the ECD, and domain stabilization in the s2 state (Figs. 6, 7, and Supplementary Figs. 8 and 9).

We next examined the region surrounding the Y184 in the 'out' position to identify structural elements facilitating the larger ECD

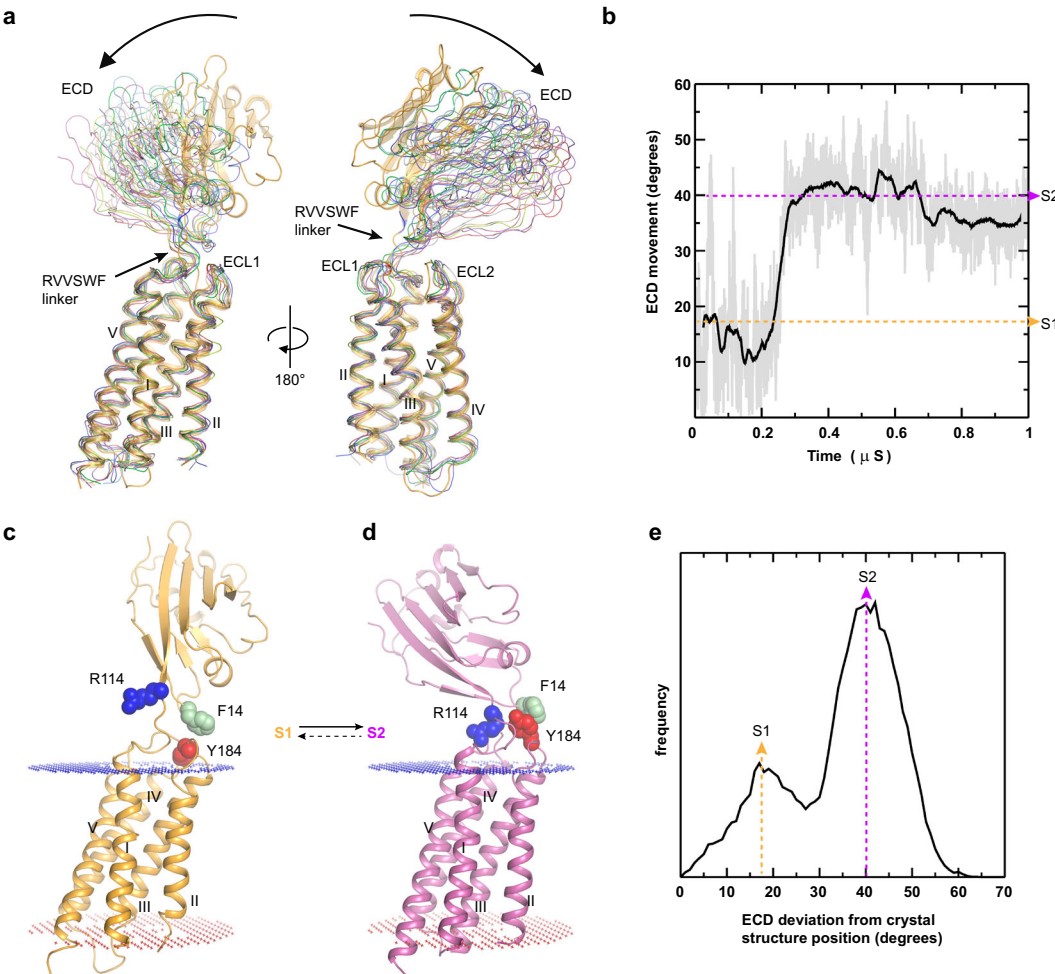

**Fig. 6 CD47 macrostates. a** Representative models of CD47 during the course of the molecular dynamics simulations taken at 0.1 μs intervals (10 models shown) showing the range of motion of CD47 ECD. The CD47 crystal structure (starting model for the simulation) is shown as a yellow cartoon and the molecular dynamics models (0.1–1 μs) are shown as thin tubes. **b** Plot showing the measured tilt (degrees) between ECD and TMD during the molecular dynamics simulations showing the transition from the s1 to s2 macrostates. The yellow (s1) and light pink (s2) dotted lines indicate the average ECD orientations in each macrostate. The gray line indicates the nanosecond fluctuations and the black line the time average over every 1 ns within the single simulation trace. **c–d** Cartoon representations of full-length CD47 models from the simulations with ECD conformations corresponding to the average macrostates s1 (yellow) and s2 (light pink). The side chains of R114, Y184, and F14 are shown as blue, red, and light green spheres respectively. **e** Plot showing the frequency distribution of the ECD positioning with respect to the TMD (measured as the angle deviation, in degrees, from the crystal structure, which is the starting molecular dynamics model, time = 0 μs). The yellow (s1) and pink (s2) dotted lines indicate the average ECD orientations in each macrostate.

hinge motion, and identified key structural rearrangements that occur in the solvent-exposed residues ([114]RVVS[117]) of the inter-domain peptide linker [114]RVVSWF[119], while W118 and F119 are maintained in the ECLR core. These residues are seen to transition from an extended conformation, to a more compacted structure during the simulations, packing in between ECL1 and ECL2, and ultimately shortening the [114]RVVSWF[119] linker to facilitate the ECD hinge motion (Fig. 7g, h). Notably, the ECD transition from s1 to s2 state, and the shortening of the [114]RVVSWF[119] linker, results in shifts of up to 6 Å from the R114 Cα position observed in the structure (R114 'out'), to a position where its side chain is docked into a negatively charged CD47 ECLR pocket (R114 'in'), between helix I and V (Fig. 7d–h).

## Discussion
In this study we determined the crystal structure of the full-length human CD47 in complex with the SIRPα blocking antibody

B6H12. CD47 is a unique 5-TM receptor with multiple biological functions and a validated drug target in cancer immuno-therapy. The data presented here reveal structural features that deepen our understanding of CD47-mediated immune 'self' recognition and signal transduction, by providing a complete structural characterization of the ECLR, and its specific interactions that bridge the TMD and ECD. The CD47 crystal structure reveals both a profound and essential role for the inter-domain peptide linker [114]RVVSWF[119], as a central element in the ECLR core. The discovery of the CD47 [117]SWF[119] loop, and the unique three-loop ECLR environment that supports the ECD on cell surfaces, provides a wealth of atomic level details to further investigate the relevance of this region for the different CD47 functions. CD47 ligands, like integrins and Tsp-1 for example, are known to require full-length CD47 for binding[27]. Such details would prove essential to fundamentally transform our understanding of cell-to-cell interactions involving CD47.

As opposed to ion channels and 7-TM GPCRs that bind diffusible molecules within the TM spanning pore, the extracellular

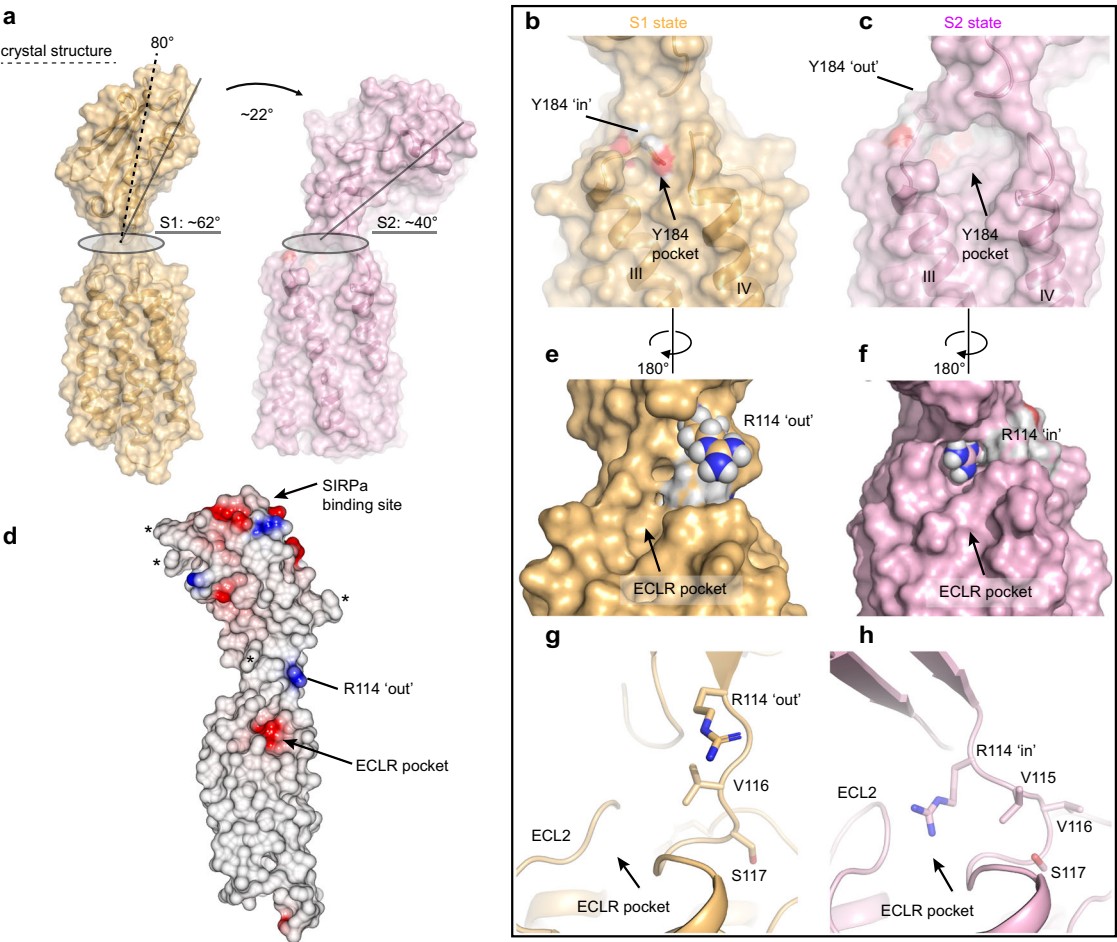

**Fig. 7 Snapshots of full-length CD47 in s1 and s2 macrostates. a** Surface representation of full-length CD47 models with ECD orientations corresponding to the average s1 (yellow) and s2 (light pink) macrostates with respect to the TMD. Differences in ECD tilt angle between s1 and s2 are indicated by black solid lines and labeled. A dotted black line indicates the ECD angle with respect to the TMD observed in the CD47[BRIL]-B6H12 crystal structure. **b**–**c** Side-by-side comparison of the Y184 large pocket between ECL1 and ECL2 showing the position of Y184 side chain in the average s1 macrostate (Y184 'in' position) and in the average s2 macrostate (Y184 'out' position). **d** Surface electrostatic potential representation of CD47[BRIL]-B6H12 crystal structure (B6H12 atoms omitted) indicating the negatively charged ECLR pocket between helix I and V, the position of R114, and the SIRPα binding site on the ECD. Asterisks indicate the position of the N-linked glycosylation sites. **e**–**f** Side-by-side comparison of R114 side chain in the average s1 macrostate (R114 'out' position) and in the average s2 macrostate (R114 'in' position) and movement of the R114 into the negatively charged ECLR pocket. **g**–**h** A cartoon representation of (**e**) and (**h**) showing the conformational rearrangement of the [114]RVVSWF[119] linker during molecular dynamics simulations. The side chain of residues are shown as sticks.

TMD core of CD47 in the crystal structure has been found to be tightly packed and inaccessible to small molecule ligands, therefore evoking an alternative mechanism of signal transduction across the lipidic membrane. Indeed, CD47 signaling has been postulated to occur via lateral association with protein partners, and the CD47 structure presented here supports this hypothesis. The observation that certain external surfaces of CD47 TMD display clear patterns of conservation provides a structural framework to probe their functional roles. In addition, a hydrogen bond network extending from the midpoint of the TM bundle, into the intracellular environment, provides the region with the potential for effector binding; the network involves helix V, the helix that presents the alternatively spliced CTD variants to the intracellular environment. In the structure presented here, in the absence of an intracellular effector, this hydrogen bond network may contribute to the stability of the 5-TM conformation.

The crystal structure presented here together with the computationally determined mechanism of ECD motion, suggests that the relative orientation of the ECD with respect to the TMD, and thus the cell membrane, is dynamic despite covalent and non-

covalent interactions between these domains. The molecular dynamics simulations suggest some of the molecular determinants of this ECD motion, whereby the ECLs can undergo conformational rearrangements, likely initiated by the 'conformational switch' of Tyr184 on ECL1, and directly mediated by the inter-domain [114]RVVSWF[119] linker. Indeed, our differential HDX data suggest that both C245 and Y184 are involved in the stabilization of the inter-domain peptide linker that includes the [117]SWF[119] loop, and also affects dynamics in the EC portion of helix I (Fig. 5 and Supplementary Fig. 7). As we expand our understanding of how CD47 contributes to the balance of adaptive and innate immunity, it is important to develop a deeper knowledge on the ECD flexibility, and if different CD47 macrostates are selectively recognized by different endogenous protein partners. In addition to the s1 ECD conformation observed in the crystal structure when bound to Fab B6H12, low-resolution single-particle Cryo-Electron Microscopy (Cryo-EM) data also suggest that the CD47[BRIL] ECD can adopt an s1-like conformation when bound to the Fab from mAb CC-90002[30] recognizing an overlapping, yet distinct epitope on CD47 ECD (Supplementary Fig. 10). Given the

similarities in the ECD positioning observed in the crystal lattice and in Cryo-EM, it appears important to further investigate whether Fab's or other binders can stabilize particular CD47 ECD conformations, or if allosteric effects mediated by ECD binding can be transmitted to the ECLR and/or TMD. This may be important for proteins that bind to large surface patches on CD47 ECD, such as Fab B6H12 (Supplementary Fig. 11).

The crystal structure of CD47$^{BRIL}$-B6H12 gives an atomic-level view of a receptor in a family of its own. The structure and mechanistic studies do not fully delineate the sequence of events that take place in CD47 signaling and ECD recognition, which will require additional high-resolution structures in complex with protein partners. It does however provide insights as to how the TM helices are oriented and how the assembly ECD-ECLR is coupled. Given the complex biology of this receptor, and its numerous physiological functions, it is expected that this single 5-TM family member evolved to acquire a high level of plasticity to meet its multiple required functions. The data provide a structural platform to understand fundamental signaling mechanisms in human biology involving CD47, such as angiogenesis, cell-to-cell communication, control of 'self' recognition, and immune responses.

## Methods

### Expression and purification of Fab's from mAb's B6H12 and CC-90002.
The B6H12 and CC-90002 mAb's were synthesized and cloned into the expression vector pTT5. Specifically, the variable heavy (VH) was synthesized as a *Homo sapiens* IgG1 heavy chain (HC), and the variable light (VL) as a *Homo sapiens* Kappa light chain (LC). All molecular biology was conducted by ATUM, and the pTT5 mammalian expression vector is licensed from the National Research Council of Canada (NRCC). B6H12 and CC-90002 were expressed transiently in ExpiCHO-S™ cells using ExpiFectamine CHO (Thermo Fisher Scientific). ExpiCHO-S cells were cultured in ExpiCHO Expression Medium (Thermo Fisher Scientific) and grown at 125 rpm (with a 25 mm orbit) at 37 °C and 5.0% CO$_2$. Prior to transfection, cells were grown to a density of $6 \times 10^6$ cells/mL in Expi-CHO™ medium. Following the manufacturer's standard protocol, 187.5 μg of HC plasmid, 187.5 μg of LC plasmid, and 2.4 ml of ExpiFectamine CHO reagent were mixed in 30 ml of cold Opti-PRO SFM (Thermo Fisher Scientific) and added to 750 mL of cells in a 3 L vented polycarbonate shake flask (Corning). The next day, 180 ml of ExpiCHO feed and 4.5 ml of ExpiCHO Enhancer were added to the ExpiFectamine-transfected cultures, mixed, and cultured for 7–9 days. Culture supernatants were clarified by centrifugation at $1865 \times g$ for 30 min and filtered using a 0.2 μm filter (Thermo Fisher Scientific). The filtered supernatant was incubated overnight with Protein-A affinity resin. The following day, the resin was captured in a gravity column, rinsed with buffer containing 50 mM Tris pH 8.0, and 150 mM NaCl until no protein was seen in the elution by Bradford reagent (BioRad). The mAb's were eluted from the Protein-A column by addition and short incubation of 1 column volume (CV) aliquots of 100 mM sodium citrate pH 3.3. Samples were diluted into a buffer containing 500 mM Tris pH 8.5 and 150 mM NaCl to neutralize the low pH of the elution buffer. Fab preparation from IgG1 mAb's were performed using the Pierce Fab Preparation kit (Thermo Fisher) according to the manufacturer's protocol. Fab samples were further purified by size exclusion chromatography, using a buffer containing 25 mM HEPES, pH 7.5, 250 mM NaCl. Protein was concentrated to 1 mg/ml$^{-1}$ and stored at 4 °C until further use. Purity, monodispersity and molecular weight of Fab samples were assessed by SDS-PAGE, analytical size-exclusion chromatography (aSEC) and mass spectrometry.

### CD47 engineering, cloning, and protein expression.
The gene of the wild type full-length human CD47 isoform 1 (residues 1–305, Uniprot accession Q08722) was codon-optimized and synthesized by Genscript for expression in *Spodoptera frugiperda* (Sf9), and then cloned into a pFastBac1 vector (Invitrogen) containing an expression cassette with a GP67 signal sequence and a 10xHis tag at the C-terminus. To obtain a full-length CD47 construct amenable to crystallography in lipidic cubic phase (LCP)[24] we performed an extensive search for a fusion protein, and precise insertion location, that would allow the formation of necessary crystal lattice contacts, particularly at the IC side of the receptor, and hence facilitate crystallization. CD47 construct screening was performed using fusion proteins known to improve crystal formation of G Protein-Coupled Receptors (GPCR's) in LCP[31], and were systematically inserted either in the ICL1, ICL2 or the C-terminal of the receptor. The final construct used in crystallization experiments contains a thermostabilized (M7W, H102I, and R106L) apocytochrome *b*562RIL from *Escherichia coli* (BRIL)[25] between residues Gly152 and Gly153 on the CD47 ICL1. Recombinant baculoviruses were generated using the Bac-to-Bac system (Invitrogen) and were used to infect Sf9 insect cells at a density of $2 \times 10^6$ cells ml$^{-1}$ at a

multiplicity of infection of 5 as previously described[32]. Infected cells were grown at 27 °C for 48 h in the presence of 10 μM kifunensine before being harvested, and the cell pellets were stored at −80 °C.

### Purification of CD47$^{BRIL}$-B6H12 Fab complex and CD47$^{BRIL}$ mutants.
Cells were disrupted by thawing the frozen cell pellets in a hypotonic buffer containing 10 mM HEPES, pH 7.5, 10 mM MgCl$_2$, 20 mM KCl, and EDTA-free protease inhibitor cocktail (Roche) with the ratio of 1 tablet per 100 ml buffer. Extensive washing of the cell membranes was performed by repeated centrifugation and dounce homogenization in the same hypotonic buffer. The cell debris was isolated by centrifugation at $40,000 \times g$ for 30 min, and then resuspended in a high osmotic buffer containing 10 mM HEPES, pH 7.5, 1 M NaCl, 10 mM MgCl$_2$, and 20 mM KCl by dounce homogenization to remove soluble and membrane-associated proteins. This step was repeated twice. The membranes were then washed by the hypotonic buffer to remove the high concentration of NaCl. The purified membranes were resuspended in 10 mM HEPES, pH 7.5, 30% (v/v) glycerol, 10 mM MgCl$_2$, 20 mM KCl, and EDTA-free complete protease inhibitor cocktail, flash-frozen with liquid nitrogen and stored at −80 °C until further use.

The purified membranes were thawed on ice and then solubilized in 50 mM HEPES, pH 7.5, 250 mM NaCl, 0.5% (w/v) *n*-dodecyl-β-D-maltopyranoside (DDM, Anatrace), 0.1% (w/v) cholesterol hemisuccinate (CHS, Sigma) at 4 °C for 3 h. The supernatant was isolated by centrifugation at $60,000 \times g$ for 30 min and incubated with TALON IMAC resin (Clontech) supplemented with 15 mM imidazole, pH 7.5 overnight at 4 °C. The resin was then washed with 15 column volumes of washing buffer 1 containing 25 mM HEPES, pH 7.5, 250 mM NaCl, 10% (v/v) glycerol, 0.05% (w/v) DDM, 0.01% (w/v) CHS, 15 mM imidazole, and followed by ten column volumes of washing buffer 2 that contains 25 mM HEPES, pH 7.5, 250 mM NaCl, 5% (v/v) glycerol, 0.05% (w/v) DDM, 0.01% (w/v) CHS and 15 mM imidazole. The receptor was then eluted with four column volumes of 25 mM HEPES, pH 7.5, 250 mM NaCl, 5% (v/v) glycerol, 0.05% (w/v) DDM, 0.01% (w/v) CHS, 220 mM imidazole. The purified Fab B6H12 at a concentration of 1 mg/ml$^{-1}$ in a buffer containing 25 mM HEPES, pH 7.5, 250 mM NaCl, 5% (v/v) glycerol, 0.05% (w/v) DDM, 0.01% (w/v) CHS was typically mixed with the CD47$^{BRIL}$ sample at 1:5 dilution to achieve at 1:1 stoichiometric ratio. Complex formation was monitored through size-exclusion chromatography (aSEC). PD MiniTrap G-25 column (GE Healthcare) was used to remove imidazole present in the CD47$^{BRIL}$-B6H12 complex sample. The protein complex was then concentrated to 40–60 mg/ml$^{-1}$ with a 100 kDa molecular weight cut-off Vivaspin centrifuge concentrator (GE healthcare) prior to crystallization experiments. For HDX-MS studies, WT and mutants were further purified by size exclusion chromatography (Superdex 200 10/300 GL, GE healthcare). The sample purity and monodispersity was analyzed by SDS-PAGE and aSEC.

### Lipidic cubic phase crystallization.
The CD47 complex samples were mixed with molten lipid (monoolein and cholesterol at a 9:1 ratio by mass) at a weight ratio of 1:1.5 (protein:lipid) using two syringes to create a LCP[24]. The mixture was dispensed onto glass sandwich plates in 40 nl drop and overlaid with 800 nl precipitant solution using a MOSQUITO LCP robot (labtech). Protein reconstitution in LCP and crystallization trials were performed at room temperature (19–22 °C). Plates were placed in an incubator (Rock Imager, Formulatrix) and imaged at 20 °C automatically following a schedule. Crystals of the CD47$^{BRIL}$-B6H12 complex appeared in 48–72 h and grew to a full size of ~$55 \times 50 \times 15$ μm$^3$ within 4 weeks in 0.1 M sodium citrate pH 5.9–6.2, 450–500 mM ammonium acetate, and 32–35% PEG400. The crystals were harvested directly from LCP using 30 and 50 μm micro mounts (M2-L19-30/50, MiTeGen), and flash-cooled in liquid nitrogen.

### Diffraction data collection and structure determination.
X-ray diffraction data were collected at the Diamond Light source at wavelength 0.986. The crystals were exposed with a 7 μm × 7 μm mini-beam for 0.2–0.4 s and 0.5° oscillation per frame. The best crystals diffracted to 3.2–3.6 Å resolution, and 10–15 frames were collected from each sample. HKL2000 (version 7.2.0)[33] was used for indexing and to obtain the unit cell parameters for each data set. The length of the smaller **b** axis of the unit cell was used as criteria to merge data sets as greater length variations were observed for this axis. Crystals with a maximal standard deviation of 2% (±SD) on the length of the **b** axis were merged in the final data set. HKL2000 was used to integrate and scale the final data set from 49 crystals. Initial phases were obtained by molecular replacement (MR) using Phaser (version 2.8.3)[34] and a molecular replacement model comprised of the CD47 ECD and Fab B6H12 from the crystal structure[26] with PDB ID 5TZU [https://doi.org/10.2210/pdb5TZU/pdb]. The MR solution contains two CD47$^{BRIL}$-B6H12 molecules in the asymmetric unit. Refinement was performed using PHENIX (version 1.14)[35] and REFMAC (version 5.8.0257)[36]. Manual model building and rebuilding of the refined coordinates were carried out in COOT (version 0.8.9.2)[37] using $|2F_o|-|F_c|$, $|F_o|-|F_c|$ and feature enhanced maps[38]. Data collection and refinement statistics are summarized in Supplementary Table 1. A total of 27 residues from the CTD of CD47$^{BRIL}$-B6H12 were flexible and could not be modeled in the structure. All molecular graphics were prepared with Pymol (version 1.8.2.2)[39].

**Analysis of TM3 contacts and structural alignment**. The structure was split into several amino acid segments and residues 114–120 were considered as the linker, 121–140 helix I, 160–178 helix II, 179–185 ECL1, 186–207 helix III, 224–242 helix IV, 243–250 ECL2 and 251–278 helix V. The CCP4i (version 7.0.078)[40] program CONTACT was used and each segment was given as an input and evaluated for proximity to TM3 (residues 186–207) at various distance cut-off between 3 and 5 Å; a distance cut-off of 3.5 Å was used to report the final contact distances in Supplementary Fig. 3. Pisa (version 2.1.1) in the CCP4i[40] and CHIMERA (version 1.14)[41] suite were used to calculate the surface buried areas. For structural alignment of CD47$^{BRIL}$-B6H12 structure with the previously determined ECD-B6H12 Fab complex structure[26], PDB ID 5TZU [https://doi.org/10.2210/pdb5TZU/pdb], we used the ECD atoms from the CD47$^{BRIL}$-B6H12 chain where the TMD was modeled (CD47 residues 1–115). Using that model as a reference, we aligned the Cα atoms of the polypeptide chain with the Cα atoms of the ECD (residues 1–115) from the ECD-B6H12 complex structure with PDB ID 5TZU [https://doi.org/10.2210/pdb5TZU/pdb] using COOT[37].

**Hydrogen/deuterium exchange mass spectrometry**. HDX-MS experiments were conducted as described previously[42]. In brief, HDX was conducted on an HDX PAL robot (LEAP Technologies, Carrboro, NC) where 5 µL of each protein construct at 1 mg/mL was diluted into 55 µL of deuterium oxide (D$_2$O) buffer (10 mM phosphate buffer, 0.05% DDM, D$_2$O, pD 7.0) to start the labeling reactions. The reactions were carried out for different time periods in the range of 20 s to 1 h in triplicate. The reaction was quenched at the end of each labeling reaction period by adding quenching buffer (100 mM phosphate buffer with 4 M guanidine hydrochloride (GdnCl) and 0.4 M tris (2-carboxylethyl) phosphine (TCEP), pH 2.5, 1:1, volume/volume), and 50 µL of the quenched sample was injected into Waters nanoACQUITY UPLC HDX Manager$^{TM}$. Proteolytic digestion was performed online using a Enzymate pepsin column, 300 Å, 5 µm, 2.1 mm × 30 mm (Waters Corp., Milford, MA, USA), at 20 °C under 100 µL/min flow rate for 3 min. The deuterated peptides were trapped and desalted for 3 min on an ACQUITY UPLC BEH C18 VanGuard Pre-column (130 Å, 1.7 µm, 2.1 mm × 5 mm) with water containing 0.1% formic acid. Peptide separation was achieved with an ACQUITY UPLC C18 BEH 1.0 × 100.0 mm column (Waters Corp.) with a 10 min gradient of 8–85% acetonitrile/water containing 0.1% formic acid operated under 40 µL/min flow rate. All the chromatographic components were held at 0.0 ± 0.1 °C. Mass spectra were obtained with a Waters Synapt G2si Q-TOF equipped with a standard ESI source (Waters Corp.). The instrument configuration was the following: capillary was 3.5 kV, sampling cone at 35 V, source temperature of 80 °C, and desolvation temperature of 175 °C. Mass spectra were acquired over an m/z range of 260–2000.

For HDX-MS data analysis, peptic peptides were identified using ProteinLynx Global Server (Waters Corp., Milford, MA, USA, version 3.0.2). The searching parameters were the following: low energy threshold was 135 counts, elevated energy threshold was 30 counts, peptide and fragment tolerances were both automatic and the primary digest reagent was non-specific. Peptide-level deuterium uptakes were calculated using Waters DynamX$^{TM}$ software (version 3.0). The % deuterium uptake was calculated using theoretical maximum number of exchangeable amide hydrogens, excluding the two N-terminal residues and proline residues in each peptide, with the consideration of ~91.7% D$_2$O in each incubation. Back-exchange assessment using fully deuterated protein was not performed in this study. The differential kinetic HDX-MS experiments were repeated at least two times and performed with a freshly purified panel of receptor constructs (CD47$^{BRIL}$ and mutants). Two independent samples were assessed for each protein construct at different times; each data-point is presented as mean values across duplicates. Error bars represent the ±SD of duplicate measurements. GraphPad Prism version 8.0.2 was used to plot all HDX data. The kinetic HDX-MS results are summarized in Supplementary Table 2.

**Molecular dynamic simulations**. The CD47 crystal structure without the Fab B6H12 was used for molecular dynamics simulations. CD47 model was prepared using the Protein Preparation Wizard module in Maestro (Schrodinger, LLC version 2020.1), and included the N-acetylglucosamines as modeled in the CD47$^{BRIL}$-B6H12 crystal structure. Protonation states of residues was determined using PropKa (version 3.1)[43]. The following are the relaxation and production simulation protocols: The prepared CD47 model was inserted into a large equilibrated palmitoyl oleolyl phosphatidylcholine (POPC) bilayer solvated with 0.15 M NaCl by: (i) inserting only the transmembrane helices portion (residues 121–140 helix I, 160–178 helix II, 186–207 helix III, 224–242 helix IV and 251–278 helix V) into the bilayer, (ii) deleting all ions and water molecules with non-hydrogen atoms within 2 Å of the protein; and (iii) deleting all lipid molecules with head-group atoms within 2 Å of the protein or lipid tail atoms within 1 Å of the protein. Salt ions were deleted to produce a system with zero net charge[44]. All simulations were run using Desmond (version 2.3) simulation engine[45], in the Schrodinger suite (version 2020.1). OPLS3 parameter set was used for the force-field[46]. The protein plus the hydrated bilayer system were relaxed using the following protocol: initially, the entire system was equilibrated for 50 ps using the NVT (constant volume and temperature thermostat; the number of particles N, volume V and temperature T, maintained throughout the simulation) Brownian dynamics at T = 10 K, with a harmonic position restraint of force constant of 5 kcal mol$^{-1}$ Å$^{-2}$ applied to all

protein heavy atoms. Next, for another 20 ps, along with the protein restraints as defined in the previous step, the membrane was also restrained along the z-axis, preventing water from entering the bilayer and the receptor, and the system equilibrated using NPT (constant pressure and temperature thermostat; the number of particles N, pressure P and temperature T, maintained throughout the simulation) Brownian dynamics, at T = 50 K. This was followed by another 100 ps of NPgT (gamma space ensemble; the number of particles N, pressure P, gamma and temperature gT maintained throughout the simulation) equilibration, at 50 K, with the same restraints. These steps allow the water to relax around the protein and the bilayer system. Next, over 150 ps, the system is heated up to 300 K while simultaneously relaxing the water barrier and lipid z-axis restraints. This is followed by another 50 ps of NVT equilibration with the positional restraints on the protein-heavy atoms preserved. Finally, through another 50 ps of NVT equilibration at T = 300 K, all positional restraints on the protein were also removed. The relaxed system was subsequently taken through three independent production simulations with the initial velocity seed randomly changed in each of the runs. Each of the three production simulations were run for a total of 1 µs in the NPgT ensemble at 300 K and 1 bar. van der Waals and short-range electrostatic interactions were cut off at 9 Å, and the pair list was updated every 12 fs[44]. Coordinates were saved every 100 ps. For all analysis, coordinates from only the production phase of the simulations were used. The calculation of the ECD position in the crystal structure of CD47$^{BRIL}$-B6H12 was performed by first orienting the receptor along a Z-axis such that center-of-mass of the TM bundle helices was set to the origin (0, 0, 0), where a plane was drawn. This step was performed using trajectory align scripts available in the Schrodinger suite (version 2020.1). Residues E123 and V274 were used as reference to define the extra and intracellular boundaries of the TMD respectively. Protein from each frame of the MD simulation was extracted and the backbone of TM helices I-V were overlaid onto the crystal structure to obtain the ECD orientation along Z-axis. ECD tilt is calculated as the angle between a vector joining the origin and the tip of the ECD defined by the center-of-mass of residues N32-V36 that form a β-hairpin loop, and the X-Y plane set at the origin (z = 0). Center-of-mass of beta-hairpin region was obtained using gmx traj analysis script in GROMACS (version 2018.1)[47], by extracting each frame of the trajectory into a PDB. Propagation of the angle between the center-of-mass of residues N32-V36 (β-hairpin loop), and the X-Y plane set at the origin (z = 0) through the molecular dynamics simulations is plotted in the graphs. The angle between the x-y plane and the vector to the center-of-mass was calculated using a C++ code available upon request. Modeling of ICL2 in the CD47 BRIL-B6H12 crystal structure was performed by manual building of the missing residues in weak electron density followed by energy minimization of the model using the Protein Preparation Wizard module in Maestro. For data analysis, the E184-K187 salt-bridge formation and Y184 movement were tracked using the simulation event analysis module in the Schrodinger suite (version 2020.1).

**Sequence alignment of CD47 from different species including viral CD47-like proteins and evolutionary conservation analysis**. All available CD47 sequences (comprised by sequences from vertebrate species and viruses) were retrieved from Uniprot (https://uniprot.org)[48] and manually inspected. To build a non-redundant amino acid sequence data set, redundant or incomplete sequences were removed. Sequence alignments were performed using Clustal Omega (www.ebi.ac.uk/Tools/msa/clustalo)[49] and T-coffee (www.tcoffee.org)[50]. Aligned sequences were rendered for visualization of sequence similarity and identity using ESPript (https://espript.ibcp.fr)[51]. For phylogenetic analyses protein sequences were aligned using AliView (version 1.26)[52] and trees were constructed using UPGMA method with 5000 bootstrapping steps as implemented in MEGAX suite (version 10.1.8)[53]. ConSurf (https://consurf.tau.ac.il)[54,55] was used to map amino acid evolutionary conservation of CD47 from different species onto the three-dimensional crystal structure of full-length CD47, and also to indicate evolutionary conservation on the sequence of human CD47.

**Cryo-Electron Microscopy sample preparation**. C-flat 1.2/1.3 holey carbon grids (Protochips) were plasma cleaned for 10 s at 25 W in Argon/Oxygen mixture. Cryo-EM grids for the full-length CD47$^{BRIL}$-Fab CC-90002 complex (prepared using the same protocol for crystallography samples described above) were prepared using a Vitrobot Mark IV system. Three milliliters of 2.0 mg/mL receptor-Fab complex, in 50 mM HEPES, pH 7.5, 250 mM NaCl, 0.05% DDM, 0.01% CHS buffer, was applied on the plasma cleaned grid. The sample was incubated on the grid for 20 s before blotting for 6 s with a blot force of zero at 4 °C in 100% relative humidity chamber of Vitrobot. Blotted grids were immediately plunged into liquid ethane cooled by liquid nitrogen.

**Cryo-Electron Microscopy data collection**. The Cryo-EM data were collected on a 300 kV FEI Titan Krios microscope equipped with a K2 direct electron detector (Gatan) and BioQuantum imaging filter (slit width 20 kV) at NanoImagine Services, San Diego, USA. Movie frames were acquired using the Leginon[56] interface for data collection. A total of 7150 movie frames at a nominal magnification of ×130,000, corresponding to 1.04 Å pixel size were acquired in the nominal defocus range of −0.3 mm to −3.1 mm. Each image was dose-fractionated to a total of 30 frames acquired over the exposure of 6 s with a total electron dose of 44.42 e$^-$/Å$^2$.

**Cryo-Electron Microscopy data processing**. 7150 dose-weighted movies were individually aligned to reduce the effects of beam-induced motion using MotionCor2[57]. The contrast transfer function for each image was estimated with program CTFFIND4[58] (version 4.0). After initial inspection, 4342 aligned and averaged images were selected for particle picking in Relion 3.0[59]. 676 particles picked using Gaussion approach as implemented in Relion 3.0 were used to generate a 2D template for automatic particle picking. 1,431,891 particles picked thus were then imported to cryoSPARC[60] (version 2.5) for further processing involving iterative 2D classification, homogeneous and heterogeneous refinements. Ab initio reconstruction was performed in cryoSPARC2.5 to obtain an initial model, which was utilized in 3D classification runs to further clean up the particle stack. The best class (structure and particles) was selected for further processing and refined in cryoSPARC2.5 using 36,786 particles, producing a 3D reconstruction with 9.8Å nominal resolution. Resolution was estimated based on the gold standard FSC = 0.143 criterion[61]. Cryo-EM density maps were visualized using Chimera[62].

**Sequence and structural database searches**. Database searches for the amino acid sequence similarity/identity were performed using the BLAST (https://blast.ncbi.nlm.nih.gov)[63] search engine. Structural database searches using either the full-length CD47 receptor or the TMD only were performed using the DALI server (http://ekhidna2.biocenter.helsinki.fi/dali)[64].

**Reporting summary**. Further information on research design is available in the Nature Research Reporting Summary linked to this article.

## Data availability

Atomic coordinates and structure factors have been deposited in the Protein Data Bank (www.rcsb.org) with accession code 7MYZ (CD47BRIL-B6H12 complex). The previously published crystal structures used in this study are available in the PDB with the accession codes 5TZU, 2JJS, and 2JJT. The source data underlying Fig. 4, Supplementary Figs. 4, 5, and 6, and Supplementary Data 1, 2, 3, and 4 are provided with this paper. The HDX-MS data generated in this study are available in the PRoteomics IDEntifications Database (PRIDE) under the accession code PXD026458. Data supporting the findings of this manuscript are available from the corresponding author upon reasonable request. Source data are provided with this paper.

## Code availability

Custom computer codes used in the molecular dynamics simulations are available upon request.

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

## Acknowledgements
We thank the i24 and i04 beamline staff members Jitka Waterman, Alex Dias, and Elizabeth Shotton for assistance with the micro focus beams at the Diamond Light Source. We thank Matt Pokross, Andres Hernandez, and Thomas Clayton for assistance with data collection and analysis. We thank Chun Luo and Mariko Riley for assistance with protein expression. We thank Jodi Muckelbauer for comments on the structure and manuscript, and Stephen R. Johnson for overseeing the molecular dynamics simulations.

## Author contributions
G.F. designed and optimized CD47 constructs for structural studies, purified and crystalized the receptor in LCP, collected and processed diffraction data, determined the structure, analyzed the data, wrote the paper, and was responsible for the overall project strategy and management. N.V. processed diffraction data, determined the structure, analyzed the data, perform phylogenetic studies, and wrote the paper. M.G. optimized and purified CD47 constructs for structural studies and crystalized the receptor in LCP. B.P. crystallized and optimized CD47 crystals. L.S.K. performed molecular dynamics simulations and analyzed the data. R.Y.-C.H. performed all HDX-MS experiments, analyzed the data, and commented on the paper. N.L. performed the cell-based ligand-binding experiments and analysis. A.S. Performed phylogenetic studies, analyzed the data, and commented on the manuscript. D.M. performed the cell-based ligand-binding experiments and analyzed the data. M.A. performed ligand-binding experiments and analyzed the data. J.J. was responsible for Ab production. H.H. optimized CD47 expression for structural studies. D.Z. oversaw the ligand-binding studies. P.C. Initiated the project and oversaw X-ray data collection and processing. H.C. and H.H. initiated and oversaw all aspects of the project.

## Competing interests
The authors declare no competing interests.
