## [Peer Review File · Nature Communications]

REVIEWER COMMENTS

Reviewer #1 (Remarks to the Author):

The authors share their results from a structural analysis of the full length CD47, including both the extracellular domain and 5 transmembrane helices, in complex with monoclonal antibody B6H12 that blocks binding to partners such as SIRP α and Tsp-1. Molecular dynamics simulations based on the crystal structure suggest that the loop connecting the ECD to the 5-TM domain in concert with a few surrounding residues can modulate the orientation of the extracellular domain. Overall these results are very interesting as they reveal the first structure of a rare 5-TM protein, and suggest how changing the orientation of the extracellular domain allows CD47 to interact with different partners. The structure should help guide experiments to dissect the details of CD47 biology.

Specific comments:

I know that extensive experiments will be required to fully understand how CD47 carries out its many functions, but it would be interesting to include some brief experimental results with mutants of Tyr184 or of the 'SWF' loop. Is signaling altered? Or in the MD simulations does mutating Tyr184 or any SWF loop residues significantly alter the dynamics of the ECD?

Merging data from 47 crystals seems challenging enough to merit more description in the supplemental material or methods section. Please comment more about how this was carried out. Were all datasets isomorphous or were many more crystals screened and only the best/most similar merged? Did you do the merging just with HKL2000 or did you utilize any specialized software such as Blend to choose which crystals to merge?

On p. 6; in 'The TMD from one unit, and the BRIL fusions could not be unambiguously resolved in the electron density maps and hence were not modelled.' Please include the residue numbers for what was and was not modelled. Also here could you please say how many N-linked glycans had electron density.

As the BRIL was added to form crystal contacts it seems odd that it is not visible in the electron density. It is difficult to tell from the manuscript and the validation report, but is there translational non-crystallographic symmetry? There appears to be only a smallish peak in the native Patterson, but the figures in the supplemental material suggest some sort of TNCS. If there is TNCS, depending on how the molecules are related it is conceivable that there could be a pseudo origin in this space

group, that if chosen incorrectly could make some density look much worse than it should (such as the missing BRIL domains and missing 5-TM). If you have not already done so, please double-check the data/model with Zanuda (in CCP4) to see if this is possible in your crystal form and if so that the origin was chosen correctly.

p. 22; What crystal screens were used? Also on this page I hope the crystals were 'flash-cooled' and not 'flash-frozen'.

p. 24-25, Is it common practice to use abbreviations for NVT, NPT, NPgT or should they be spelled out or defined at first use?

Please include a figure in the supplemental material showing some section of the model with associated electron density.

In the PDB validation report, for R and Rfree calculated by the DCC, the Rfree is slightly lower than R and significantly different from author reported Rfree. Please resolve this issue and provide a valid PDB code, not XXXX.

Reviewer #2 (Remarks to the Author):

Fenalti et al. describe the crystal structure of full length CD47 at 3.4 Å resolution in lipidic mesophases. Although the crystal structure of the CD47 ectodomain had previously been solved in complex with the B6H12 Fab (PDB ID: 5TZU), this is the first description of the architecture of the 5-TM bundle of CD47. The authors identify residues W118 and F119, located in the loop that connect the ectodomain to the TMD as key residues for receptor stabilization. In addition, they perform molecular dynamic simulations and report that CD47 adopts two conformations mainly driven by the position of residue Y184. This work constitutes an important new structure of CD47.

Major concerns:

1. Although the structural information is interesting on its own, and the structure is well described, the claims made with regards to biological implications of the structural findings are highly limited. For example: 1) Title: “Structure of the human 5-transmembrane receptor CD47 reveals insights into immune ‘self’ recognition”. There is little to no experimental insight into immune recognition. 2) “The insertion of the 117SWF119 loop residues into the center of the ECLR, together with the C15-C245 inter-domain disulfide bond are the two ‘anchor’ points tethering the CD47 ECD to the ECLR. They act in concert to maintain the structural integrity of the ECLR core and optimal positioning of the ECD on the cell surface, critically required for the different biological functions of the receptor”. No experimental insights are provided for these different biological functions. 3) “The discovery of the CD47 117SWF119 loop, and the unique three loop ECLR environment that supports the ECD on cell surfaces, provides a wealth of atomic level details to better understand the binding mechanisms to numerous other endogenous ligands”. No experimental data is provided to support such binding mechanisms to ligands. It is this Reviewer’s opinion that binding experiments are needed to compare the affinity of SIRP α or other ligands to WT CD47 and the binding affinity to mutated versions of CD47 in the SWF loop and in Y184 – the critical residues uncovered in this structure.

2. Molecular dynamic simulations were performed in the presence of POPE. The choice of this lipid should be explained and the implication on having a negative membrane curvature in the observed flexibility of CD47 should be considered and discussed. In addition, the biological relevance of having two conformations of CD47 should be tested with additional experiments or at least discussed.

3. It has been shown that CD47 clusters in lipid rafts in non-apoptotic and hence increasing binding avidity to SIRP α to prevents phagocytosis. Lipid rafts are rich in cholesterol and glycosphingolipids. Does the crystal structure of the TMD provide any insights into a preferred partitioning of CD47 into this ordered lipid domains? Structure-function evidence to this effect would be informative.

Minor comments:

1. It is unfortunate how some of the points mentioned in the Results section refer to representations spread over several figures e.g. “[...] form a tilted (45°) edge-to-face aromatic interaction between them (Fig. 1b, c, Fig. 2a and Fig. 3a).” It would be easier for the reader if points in a given section were organized or described concisely within one figure, and that the next figure focused on different points.

2. Several terms need to be defined such as: BRIL in the results or EC and IC in Fig 1.

3. “80.1° angle with respect to the TMD” should be rounded to “80° angle with respect to the TMD”.

4. At this resolution, the buried surface area values should be rounded to the nearest unit: e.g. “The hydrophobic aromatic side chains of W118 and F119 on the 117SWF119 loop are completely buried (319.1Å²)” should be 319 Å².

5. Please provide a rationale for the residue range selected for the rmsd calculation to describe the similarity between the full-length structure of CD47 and the previously described CD47 ectodomain structure.

6. In the author contribution section, it mentions: “M.A. performed ligand binding experiments”. No ligand binding experiments were presented in the manuscript, although such data would be very welcome to complement the structure.

7. Several figures show the side chain of residues that are not labeled. These should be either labeled or the side chains should not be shown.

Reviewer #3 (Remarks to the Author):

Summary

The manuscript by Fenalti et al. presents (to my understanding) the first full-length structure of any 5-TM human receptor—in this case CD47, a receptor of importance to the immune system. The ECD of CD47 is displayed upon the surface of many cells and binds a receptor on macrophages (SIRP-alpha), signaling the macrophage to ‘not eat’ the circulating, CD47-displaying cell. Certain cancer cells may overexposes CD47, enabling them to evade the immune system, and thus pharmaceuticals that block the interaction between CD47 and SIRP-alpha hold potential as cancer therapeutics. Moreover, CD47 can signal in its own right intracellularly, stimulating myriad downstream signaling cascades.

In this manuscript, the authors present the structure of CD47 bound to the inhibitory antibody, B6H12. They find that a key linker, termed the SWF loop, allows the ECD to contact the ECL interface. Evolutionary and sequence analysis suggests that, while the SIRP-alpha binding surface is highly conserved, other surfaces on the ECD are not. Fascinatingly, even the evolutionary distance CD47-like receptors in viruses share a small number of highly-conserved residues, including 118W and 119F. Molecular dynamics simulations reveal that the ECD is highly mobile and can tilt along a hinge-like axis.

My primary reaction is that this is an interesting paper from a receptor biology and cancer therapeutics standpoint. My only concern is that the paper lacks few insights into how the structure of CD47 determines its function. The paper contains no new mutagenesis, binding, or other downstream (functional) assays, making it difficult to assess the importance of certain structural and simulation-based observations.

Major comments:

1. Aside from certain references to previous mutagenesis studies, little is done in this paper to map the important functional interfaces and/or surface patches on the ECD of CD47. I'm not sure the extent to which this could be done, but could the authors at least add some discussion on potential ways in which the structure sheds light on the types of interactions it can form with not just with SIRP-alpha but also other extracellular binding partners? Is there some aspect of the ECD surface that is especially druggable?

2. Analogously to the first point, what, if anything, can the authors infer about how binding on the extracellular surface affects the conformational dynamics of the intracellular side of the receptor? Is there any evolutionary relationship to other membrane-bound human receptors? Could be particularly useful on p. 9–11 when the authors are describing the global architecture of the receptor's TM domain

3. The authors use MD to show that the ECD's hinge-like motion contributes to it adopting two different orientations with respect to the membrane plane, here called states s1 and s2. They refer to these states as 'metastable' — but I think that in order to refer to these states as metastable, it is necessary to demonstrate that the ECD can transition back and forth between the two states. That is, it is not enough to say that a simulation started out in state s1 and sometimes transitioned to state s2. In that case, s1 might not represent a well-populated state in the true conformational landscape, since it might be favored by crystallization conditions (for example); in that case, s2 isomer representative of the native eCD orientation. If so, then do certain contacts within or outside of the crystallographic asymmetric unit (or with the Fab) stabilize this particular s1 ECD orientation? Moreover, the authors should report information for each simulation replicate and not just for the simulations in aggregate. This should be straightforward to do since they've only performed three simulations.

4. It is difficult to know what implications the simulation observations have for how CD47 functions. What seems to be lacking from this manuscript is a way to assess the impact of a mutation (say in Y184) on function—where function includes both binding to the SIRP-alpha receptor, as well as any

other downstream interactions that might occur in cells. I realize it might be difficult now during covid-19 for the authors to return to the lab and carry out some sort of functional assay to ascertain the significance of residues like Y184. If that's the case, then perhaps the authors could carry out in-silico mutagenesis studies, and in vitro binding assays, to determine how mutation of Y184 affects ECD mobility in simulation, and to assess its impact on receptor binding.

Minor comments:

1. It is quite hard to make out the disulfide bridges in Fig. 1B. Perhaps color the sulfurs by element?
2. p. 12, line 253: should 'levered' be 'leveraged'?
3. Do the simulations reveal any interesting change on the intracellular side of the receptor relative to the initial, crystallographic conformation?

REVIEWER COMMENTS

Reviewer #1 (Remarks to the Author):

The authors share their results from a structural analysis of the full length CD47, including both the extracellular domain and 5 transmembrane helices, in complex with monoclonal antibody B6H12 that blocks binding to partners such as SIRP α and Tsp-1. Molecular dynamics simulations based on the crystal structure suggest that the loop connecting the ECD to the 5-TM domain in concert with a few surrounding residues can modulate the orientation of the extracellular domain. Overall, these results are very interesting as they reveal the first structure of a rare 5-TM protein and suggest how changing the orientation of the extracellular domain allows CD47 to interact with different partners. The structure should help guide experiments to dissect the details of CD47 biology.

Specific comments:

I know that extensive experiments will be required to fully understand how CD47 carries out its many functions, but it would be interesting to include some brief experimental results with mutants of Tyr184 or of the 'SWF' loop. Is signaling altered? Or in the MD simulations does mutating Tyr184 or any SWF loop residues significantly alter the dynamics of the ECD?

We thank the reviewer for these suggestions which improved our manuscript. As requested, we included new experimental data from kinetic hydrogen-deuterium exchange mass spectrometry (HDX MS) experiments using WT CD47 and mutants. The data provides new insights into the dynamic profile of the extra cellular loop region (ECLR) and portions of the TMD. The HDX data delineates important roles for residues Y184, W118 and C245 and their specific impact on the conformational dynamics of the inter-domain ¹¹⁴RVVSWF¹¹⁹ linker region, and on the TMD of the receptor (EC portion of helix I). These findings are consistent with the notion that the inter-domain ¹¹⁴RVVSWF¹¹⁹ 'hinge' mobility is incrementally affected by Y184F, Y184A and

C245S mutants (in the ECL1 and ECL2 respectively). We have included this experimental data under a new sub-heading in the results section namely “*ECL residues controlling ¹¹⁴RVVSWF¹¹⁹ dynamics*” (page 14). We could not pursue CD47 signaling studies, those would be difficult to complete during lab restrictions imposed by the COVID-19 pandemic. However, we performed additional CD47-SIRP α binding studies using CD47 WT and mutants (Fig. 1, see methods below). These brief experiments are shown in this letter but were not included in the manuscript. We did not pursue *in silico* studies of CD47 mutants.

Merging data from 47 crystals seems challenging enough to merit more description in the supplemental material or methods section. Please comment more about how this was carried out. Were all datasets isomorphous or were many more crystals screened and only the best/most similar merged? Did you do the merging just with HKL2000 or did you utilize any specialized software such as Blend to choose which crystals to merge?

We thank the reviewer for pointing this out. Although crystals of the complex CD47^{BRIL}-B6H12 in LCP were not very isomorphous in nature, the biggest challenge in this program was to obtain enough crystals that diffracted to a moderate resolution to generate a complete dataset. We estimate that ~1500 crystals were screened and only a very low percentage of crystals diffracted to a resolution between 3.2Å-3.6Å. Only those crystals that diffracted to a resolution better than 3.6Å and had a maximal standard deviation of 2% (\pm SD) on the length of the smaller *b* axis of the unit cell were merged in the final data set. We have used HKL2000 for merging and scaling selected data sets and have expanded the methods section under ‘*diffraction data collection and structure determination*’ page 25 to further clarify our crystal merging and filtering strategy.

On p. 6; in ‘The TMD from one unit, and the BRIL fusions could not be unambiguously resolved in the electron density maps and hence were not modelled.’ Please include the residue numbers for what was and was not modelled. Also, here could you please say how many N-linked glycans had electron density.

As requested by the reviewer, we have now included a description of the regions of the CD47 structure that have been modeled or not on page 6 under the results sub section ‘*Overall*

architecture of the CD47^{BRIL}-B6H12 complex'. We added a sentence to specify how many N-linked glycans had density on page 7 of the manuscript.

As the BRIL was added to form crystal contacts it seems odd that it is not visible in the electron density. It is difficult to tell from the manuscript and the validation report, but is there translational non-crystallographic symmetry? There appears to be only a smallish peak in the native Patterson, but the figures in the supplemental material suggest some sort of TNCS. If there is TNCS, depending on how the molecules are related it is conceivable that there could be a pseudo origin in this space group, that if chosen incorrectly could make some density look much worse than it should (such as the missing BRIL domains and missing 5-TM). If you have not already done so, please double-check the data/model with Zanuda (in CCP4) to see if this is possible in your crystal form and if so that the origin was chosen correctly.

We thank the reviewer for this comment and suggestion. Although BRIL was used as a fusion protein in the full length CD47 to facilitate formation of crystal lattice contacts in LCP, in the case of CD47^{BRIL}-B6H12 crystal structure, the larger Fab from B6H12 mediated almost all crystal lattice contacts. As a result, BRIL is not involved in specific contacts that can 'lock' the fusion protein in a specific orientation in the lattice, therefore contributing to flexibility and lack of electron density for the protein. We have checked our data and model using Zanuda¹ as suggested by the reviewer and confirmed the correct assignment of both origin and space group. We refined the crystal structure of CD47^{BRIL}-B6H12 to a final R_{free} of 28.1. Typically, when a mistake is made in the assignment, it becomes apparent when the R_{free} ceases to drop below ~ 0.39 , and model building/refinement don't improve refinement statistics. This does not seem to be the case for the CD47^{BRIL}-B6H12 structure. It is worthwhile mentioning that prior to this work, we determined the crystal structure of CD47^{BRIL} in complex with a Fab recognizing a different epitope on the ECD (CC-90002). Unfortunately, although the Fabs, CD47 ECDs and BRILs were visible in electron density maps, the TMD's were not. We were also unable to determine the single particle Cryo-EM structure of the full length CD47 (**Fig.2**), suggesting that there is an intrinsic level of conformational variability for this receptor that is refractory to structural characterization.

p. 22; What crystal screens were used? Also, on this page I hope the crystals were 'flash-cooled' and not 'flash-frozen'.

For broad screening of crystallization conditions, we made custom crystallization screen matrices containing different salts from the StockOptions Salt kit (Hampton Research) at 100mM and 400mM concentration, PEG400 (20%-35%), and 100mM of a specific buffer (Tris, Hepes or Citrate) to cover a pH range between 5.5-8. A typical 96 well crystallization screen would contain 48 different salts at a concentration of 100mM and 400mM, a fixed percentage of the precipitant PEG400, and 100mM of the desired buffer. We have corrected the sentence in the **Materials and Methods** section which now reads "... and flash-cooled in liquid nitrogen."

p. 24-25, Is it common practice to use abbreviations for NVT, NPT, NPgT or should they be spelled out or defined at first use?

We have now addressed this issue and spelled out NVT, NPT, NPgT in the **Molecular dynamic simulations** section of **Materials and Methods**.

Please include a figure in the supplemental material showing some section of the model with associated electron density.

We included a new Supplementary Fig. 2 showing the electron density for different sections of the complex structure.

In the PDB validation report, for R and Rfree calculated by the DCC, the Rfree is slightly lower than R and significantly different from author reported Rfree. Please resolve this issue and provide a valid PDB code, not XXXX.

We thank the reviewer for pointing that out. We have resolved these inconsistencies in the PDB validation report and provided a PDB code (PDB ID 7MYZ).

Reviewer #2 (Remarks to the Author):

Fenalti et al. describe the crystal structure of full length CD47 at 3.4Å resolution in lipidic mesophases. Although the crystal structure of the CD47 ectodomain had previously been solved in complex with the B6H12 Fab (PDB ID: 5TZU), this is the first description of the architecture of the 5-TM bundle of CD47. The authors identify residues W118 and F119, located in the loop that connect the ectodomain to the TMD as key residues for receptor stabilization. In addition, they perform molecular dynamic simulations and report that CD47 adopts two conformations mainly driven by the position of residue Y184. This work constitutes an important new structure of CD47.

Major concerns:

1. Although the structural information is interesting on its own, and the structure is well described, the claims made with regards to biological implications of the structural findings are highly limited.

For example:

1) Title: “Structure of the human 5-transmembrane receptor CD47 reveals insights into immune ‘self’ recognition”. There is little to no experimental insight into immune recognition.

We thank the reviewer for this comment and appreciate the fact that extensive studies would be required to fully understand the mechanism of immune recognition between full length SIRPα and full length CD47. To address the reviewer’s request we simplified the manuscript title to read ‘**Structure of the human ‘marker of self’ 5-transmembrane receptor CD47**’.

2) “*The insertion of the 117SWF119 loop residues into the center of the ECLR, together with the C15-C245 inter-domain disulfide bond are the two ‘anchor’ points tethering the CD47 ECD to the ECLR. They act in concert to maintain the structural integrity of the ECLR core*”

and optimal positioning of the ECD on the cell surface, critically required for the different biological functions of the receptor”. No experimental insights are provided for these different biological functions.

This is an excellent point brought by the reviewer and we omitted the second sentence of the above-mentioned text. It now reads ...“The insertion of the ¹¹⁷SWF¹¹⁹ loop residues into the center of the ECLR, together with the C15-C245 inter-domain disulfide bond are the two ‘anchor’ points tethering the CD47 ECD to the ECLR. ~~They act in concert to maintain the structural integrity of the ECLR core and optimal positioning of the ECD on the cell surface. critically required for the different biological functions of the receptor.~~”... To shed some light into the conformational dynamics of the CD47 ECLR, the interplay between the inter-domain disulfide bond and inter-domain ¹¹⁴RVVSWF¹¹⁹ linker, and how these may be recruited for receptor function, we complemented our structural work with a kinetic hydrogen-deuterium exchange mass spectrometry (HDX-MS) study. Experiments were performed with wild type (WT) CD47 and mutants, and the data is included in the **Results** section under the new subheading *ECL residues controlling ¹¹⁴RVVSWF¹¹⁹ linker dynamics*. The HDX results showed that removal of the inter-domain disulfide bond (C245S mutant) have a major effect on the conformational stability of the inter-domain ¹¹⁴RVVSWF¹¹⁹ linker. In addition to the interpretation of the crystal structure, the HDX data suggests an intimate functional relationship between the inter-domain disulfide bond and the linker, where a properly formed disulfide bond between domains stabilizes the conformation of the ¹¹⁴RVVSWF¹¹⁹ linker. Mutation at position Y184 also leads to the same specific effects on the dynamics of the ¹¹⁴RVVSWF¹¹⁹ linker. Normal binding to SIRPα, one of the many biological functions of the receptor, is impaired by loss of this disulfide bond (**Fig.1** below) and consequent increased conformational dynamics of the ¹¹⁴RVVSWF¹¹⁹ linker shown here by HDX analysis. Mutation of Y184 also has a significant HDX effect on the ¹¹⁴RVVSWF¹¹⁹ linker connecting to the ECD, as predicted from analysis of the crystal structure and MD simulations.

3) “The discovery of the CD47 ¹¹⁷SWF¹¹⁹ loop, and the unique three loop ECLR environment that supports the ECD on cell surfaces, provides a wealth of atomic level details to better understand the binding mechanisms to numerous other endogenous ligands”. No experimental data is provided to support such binding mechanisms to ligands. It is this

Reviewer's opinion that binding experiments are needed to compare the affinity of SIRP α or other ligands to WT CD47 and the binding affinity to mutated versions of CD47 in the SWF loop and in Y184 – the critical residues uncovered in this structure.

We thank the reviewer for this comment. This sentence was changed to more appropriately indicate that further studies would benefit from the availability of this CD47 structure crystal structure, and it now reads ...“The discovery of the CD47¹¹⁷SWF¹¹⁹ loop, and the unique three loop ECLR environment that supports the ECD on cell surfaces, provides a wealth of atomic level details to further investigate the relevance of this region for the different CD47 functions”... The binding and signaling mechanisms of different CD47 partners (*in cis* or *in trans*) are certainly complex and likely have different structural requirements for specific receptor functions. As requested, we performed some binding studies using a CD47 *null* cell line overexpressing WT CD47 and mutants, and measured binding to recombinant human SIRP α (D1 domain, R&D Systems, dimeric) (**Fig.1**). Consistent with the literature², the C245S mutant lost SIRP α affinity compared to WT, yet the Y184 mutants had no changes in affinities compared to the WT. These results demonstrate a positive correlation to the observations in the HDX study, showing that C245S has a major effect in the conformational dynamics of the ¹¹⁴RVVSWF¹¹⁹ linker that is associated to loss of SIRP α binding function. While the magnitude of this HDX effect is partially induced by Y184 mutants, its effects on SIRP α binding are not detected. This assay method (see methods below) is robust and has been used by different groups in the field, despite using dimeric SIRP α which increases affinity to the nM range as it creates an avidity effect, as opposed to the known ~1 μ M affinity between CD47-SIRP α that is more representative of the physiological conditions. We are concern about the sensitivity of this assay to detect more subtle differences in binding to SIRP α . A more comprehensive structure-function study would be required to fully understand the impact of Y184 on the SIRP α signaling axis, or in other functions of the receptor. Such in depth studies are beyond the scope of this manuscript and we want to be careful making statements about the function of this residue based on limited data. Our binding data is shown below but not included in the manuscript.

Fig. 1. Binding of CD47 variants to dimeric SIRP α

- 4) Molecular dynamic simulations were performed in the presence of POPE. The choice of this lipid should be explained and the implication on having a negative membrane curvature in the observed flexibility of CD47 should be considered and discussed. In addition, the biological relevance of having two conformations of CD47 should be tested with additional experiments or at least discussed.

We want to thank the reviewer for catching this and apologize for the mistake. We used palmitoyl oleoyl phosphatidyl choline (POPC) and not the charged POPE as stated in the previous version of the manuscript. The manuscript has been updated accordingly.

- 5) It has been shown that CD47 clusters in lipid rafts in non-apoptotic and hence increasing binding avidity to SIRP α to prevents phagocytosis. Lipid rafts are rich in cholesterol and glycosphingolipids. Does the crystal structure of the TMD provide any insights into a preferred partitioning of CD47 into this ordered lipid domains? Structure-function evidence to this effect would be informative.

We thank the reviewer for pointing out the importance of cholesterol. We agree that knowledge of such binding sites would be valuable. Our data shows no electron density consistent with the presence of cholesterol in this structure, and unfortunately discussions on the putative cholesterol binding sites would be speculative at this point.

Minor comments:

1. It is unfortunate how some of the points mentioned in the Results section refer to representations spread over several figures e.g. “[...] form a tilted (45°) edge-to-face aromatic interaction between them (Fig. 1b, c, Fig. 2a and Fig. 3a).” It would be easier for the reader if points in a given section were organized or described concisely within one figure, and that the next figure focused on different points.

We thank the reviewer for the feedback on the arrangement of figures. It was our intention to have each figure focusing on different features of the structure as suggested by the reviewer, for example: Fig. 1 is an overall view of the complex, with a close view of the ECLR packing and top and bottom views of the TMD; Fig. 2 is focused on the hydrogen bond interactions within the ECLR and close view of the TM helices; Fig. 3 highlights the hydrophobic core of the TMD; Fig 4 shows the amino acid conservation and so on. We agree that some of the structural details appear in multiple figures, but we aimed to thoroughly cite these features of the receptor, and this may give the reader an opportunity to visualize the same structural features in slightly different orientations, or in a different context (e.g. amino acid conservation and type of atomic interactions).

2. Several terms need to be defined such as: BRIL in the results or EC and IC in Fig 1.

We thank the reviewer for bring this to our attention. We had previously defined BRIL in the methods section but not in the main text **Results** section. We have now added the BRIL description in the **Results** section as suggested and spelled out ‘extracellular’ and ‘intracellular space’ in Fig. 1.

3. “80.1° angle with respect to the TMD” should be rounded to “80° angle with respect to the TMD”.

We have addressed the above request in the new version of the manuscript.

4. At this resolution, the buried surface area values should be rounded to the nearest unit: e.g. “The hydrophobic aromatic side chains of W118 and F119 on the 117SWF119 loop are completely buried (319.Å²)” should be 319Å².

We have used the rounded values in the new version of the manuscript as suggested by the reviewer.

5. Please provide a rationale for the residue range selected for the rmsd calculation to describe the similarity between the full-length structure of CD47 and the previously described CD47 ectodomain structure.

For structural alignment of CD47^{BRIL}-B6H12 structure with the previously determined ECD-B6H12 Fab structure (PDB ID 5TZU), we used the ECD atoms from the CD47^{BRIL}-B6H12 chain where the TMD was modeled (CD47 residues 1-115). Using that model as a reference, we aligned the C α atoms of the polypeptide chain with the C α atoms of the ECD (residues 1-115) from the ECD-B6H12 complex structure 5TZU using COOT³. We included a description of this procedure on page 27 of the manuscript.

6. In the author contribution section, it mentions: “M.A. performed ligand binding experiments”. No ligand binding experiments were presented in the manuscript, although such data would be very welcome to complement the structure.

M.A. performed binding experiments early in the project and confirmed the binding of Fab fragments to the lead CD47 crystal constructs. These results are not shown in the manuscript.

7. Several figures show the side chain of residues that are not labeled. These should be either labeled or the side chains should not be shown.

We thank the reviewer for this comment. We have made the requested corrections in Fig. 2, Fig. 3, Fig. 4, Fig.7 and Supplementary Fig. 3.

Reviewer #3 (Remarks to the Author):

Summary

The manuscript by Fenalti et al. presents (to my understanding) the first full-length structure of any 5-TM human receptor—in this case CD47, a receptor of importance to the immune system. The ECD of CD47 is displayed upon the surface of many cells and binds a receptor on macrophages (SIRP-alpha), signaling the macrophage to ‘not eat’ the circulating, CD47-displaying cell. Certain cancer cells may overexpress CD47, enabling them to evade the immune system, and thus pharmaceuticals that block the interaction between CD47 and SIRP-alpha hold potential as cancer therapeutics. Moreover, CD47 can signal in its own right intracellularly, stimulating myriad downstream signaling cascades.

In this manuscript, the authors present the structure of CD47 bound to the inhibitory antibody, B6H12. They find that a key linker, termed the SWF loop, allows the ECD to contact the ECL interface. Evolutionary and sequence analysis suggests that, while the SIRP-alpha binding surface is highly conserved, other surfaces on the ECD are not. Fascinatingly, even the evolutionary distance CD47-like receptors in viruses share a small number of highly-conserved residues, including 118W and 119F. Molecular dynamics simulations reveal that the ECD is highly mobile and can tilt along a hinge-like axis.

My primary reaction is that this is an interesting paper from a receptor biology and cancer therapeutics standpoint. My only concern is that the paper lacks few insights into how the structure of CD47 determines its function. The paper contains no new mutagenesis, binding, or other downstream (functional) assays, making it difficult to assess the importance of certain structural and simulation-based observations.

Major comments:

1. Aside from certain references to previous mutagenesis studies, little is done in this paper to map the important functional interfaces and/or surface patches on the ECD of CD47. I’m not

sure the extent to which this could be done, but could the authors at least add some discussion on potential ways in which the structure sheds light on the types of interactions it can form with not just with SIRP-alpha but also other extracellular binding partners? Is there some aspect of the ECD surface that is especially druggable?

We appreciate the thoughtful comments of the reviewer. We agree that it would be critical to understand how the full length CD47 structure determines one, or more of its multiple functions. Such studies are challenging and would require a large body of work, and hopefully the crystal structure presented here will aid better understanding of the signaling of this receptor. To complement our work on the full length CD47 structure, we have included additional kinetic hydrogen-deuterium exchange mass spectrometry (HDX-MS) data in the revised version of the manuscript, which was performed with WT CD47 and mutants. The data reveals a specific effect of residues on the ECL's in the conformational dynamics of the ¹¹⁴RVVSWF¹¹⁹ linker, which in the case of C245S mutant, is associated with impaired CD47 binding to SIRP α (as reported by Rebres *et al.*, and now confirmed by us, **Fig.1**). To the best of our knowledge, the functional interfaces on CD47 that have been structurally characterized are for SIRP α , and binding sites for different Fab's; these sites cluster around the SIRP α binding site and are further away from the CD47 ECLR. We have not commented on the manuscript the types of interactions CD47 may be able to make other than with SIRP α as we fear these may be inaccurate, and ideally a structure *in cis* with a CD47 binding partner (e.g. integrin) would be required. The question about possible druggable pockets is interesting, but without some additional experiments it is difficult to determine if there are any potential druggable pockets on the ECD surface. There may be cryptic ECLR pockets that can form because of movements on the ECD and/or ECLR regions; pockets such as the ones formed by the 'in' or 'out' position of Y184, or R114.

2. Analogously to the first point, what, if anything, can the authors infer about how binding on the extracellular surface affects the conformational dynamics of the intracellular side of the receptor? Is there any evolutionary relationship to other membrane-bound human receptors? Could be particularly useful on p. 9–11 when the authors are describing the global architecture of the receptor's TM domain

This is a very interesting point raised by the reviewer. We are also very interested in how the signal is transmitted from the extracellular space, through the ECLR and TMD and into the intracellular effector domain. Based on our data, it's difficult to define such a mechanism, however, there are some interesting additional structural features that may relate to how the C-terminal domain is presented in the intracellular space. If the interactions of the intracellular hydrogen bond network around H206 can be broken, instability in the intracellular portion of the connecting helices can be further exacerbated. According to our HDX data of the C-terminal of CD47, the intracellular portion of helices III and V are indeed in fast exchange. Further, the effector C-terminal domain (which could not be modeled in the structure) showed the highest levels of HDX uptake of all receptor. It is difficult to ascertain if any of these features are important for either *in cis* association of the TMD and integrins or *in trans* through the ECD without additional complex structures. Given that this is the only 5-TM receptor in the human genome, and it displays a novel fold, it was not possible to define a functional evolutionary relationship to other membrane proteins.

3. The authors use MD to show that the ECD's hinge-like motion contributes to it adopting two different orientations with respect to the membrane plane, here called states s1 and s2. They refer to these states as 'metastable' — but I think that in order to refer to these states as metastable, it is necessary to demonstrate that the ECD can transition back and forth between the two states. That is, it is not enough to say that a simulation started out in state s1 and sometimes transitioned to state s2. In that case, s1 might not represent a well-populated state in the true conformational landscape, since it might be favored by crystallization conditions (for example); in that case, s2 isomer representative of the native ECD orientation. If so, then do certain contacts within or outside of the crystallographic asymmetric unit (or with the Fab) stabilize this particular s1 ECD orientation? Moreover, the authors should report information for each simulation replicate and not just for the simulations in aggregate. This should be straightforward to do since they've only performed three simulations.

We thank the reviewer for raising this interesting point. We agree that showing data across all simulation replicates helps with clearer data

Fig. 2. a) 10Å Cryo-EM reconstruction of the Fab 90002/CD47^{BRIL} complex. b) Exemplary class average of the Fab 90002/CD47^{BRIL} complex.

interpretation and we have added this data in the **Supplementary Figure 9**. In each of the simulation replicates, the ECD transitions to the S2 state at different simulation times (simulation 1 : $\sim 0.2\mu\text{s}$, simulation 2 : $\sim 0.6\mu\text{s}$ and simulation 3: $\sim 0.1\mu\text{s}$), and in each of these cases, the ECD movement correlates with the movement of Y184 from the ECLR pocket. This suggests that the ECD orientation identified as S1 is unlikely due to crystallization conditions, but a more stable state that the ECD occupies. The ECD likely continues to sample S1 state conformation until an event such as the Y184 movement out of TM pocket triggers its transition to S2. However, as pointed by the reviewer the ECD position doesn't return to the s1 state after transitioning to s2, and so we termed them 'macro-states' as opposed to meta-stable states, as used in the previous version of the manuscript. Additionally, we have collected $\sim 10\text{\AA}$ single particle Cryo-EM data for CD47^{BRIL} in complex with the Fab from the clinical molecule CC-90002. Despite the low resolution of the data and the fact that the receptor TMD could not be visualized in the electron density maps, the micelle provides a reference plane mimicking the outer cell membranes. By independently determining the structure of the soluble CD47 ECD in complex with the Fab 90002 and fitting the model into the Cryo-EM maps we noted that the CD47 ECD assumes a S1 like orientation. Altogether, these results indicate that the ECD orientations observed in the crystal structure and through MD are representative of two macro-states that the receptor ECD occupies and are unlikely artifacts of crystallization or simulation conditions.

4. It is difficult to know what implications the simulation observations have for how CD47 functions. What seems to be lacking from this manuscript is a way to assess the impact of a mutation (say in Y184) on function—where function includes both binding to the SIRP-alpha receptor, as well as any other downstream interactions that might occur in cells. I realize it might be difficult now during covid-19 for the authors to return to the lab and carry out some sort of functional assay to ascertain the significance of residues like Y184. If that's the case, then perhaps the authors could carry out in-silico mutagenesis studies, and in vitro binding assays, to determine how mutation of Y184 affects ECD mobility in simulation, and to assess its impact on receptor binding.

In the revised version of the manuscript we describe a kinetic HDX-MS study on the WT and mutants. The data suggests that both C245 and Y184 have significant structural effects on the

conformational dynamics of the ¹¹⁴RVVSWF¹¹⁹ linker, and in the case of C245, the greater impact in the linker seems to correlate with impaired SIRP α binding. We performed some binding experiments with Y184 mutants (**see above, Fig.1**) but further studies are required to understand how they affect SIRP α binding and signaling, preferably using a highly sensitive, cell-to-cell signaling assay.

Minor comments:

1. It is quite hard to make out the disulfide bridges in Fig. 1B. Perhaps color the sulfurs by element?

We have updated the Fig.1 to clearly show the disulfide bridges.

2. p. 12, line 253: should 'levered' be 'leveraged'?

We have corrected this typo.

3. Do the simulations reveal any interesting change on the intracellular side of the receptor relative to the initial, crystallographic conformation?

We noted that the intracellular portion of helix V (last two helical turns) tend move as a rigid body which may be facilitated by the presence of a glycine residue near the C-terminal of helix V.

Ligand Binding assay protocol

SIRP α binding to CD47 mutants overexpressed into HEK293T cells.

To test SIRP α binding to CD47 mutants, 1 million cells of CD47 KO HEK293T cells (Abcam ab266324) were plated into T-75 flasks, in 10 mL of DMEM media containing 10% FBS. 24 hours later, cells were

transfected with 6 μg CD47 construct DNA (WT or mutants) mixed with 36 μl Fugene HD (Promega E2311) transfection reagent. 48hrs post transfection, cells were lifted, counted, and plated 100K per well in round bottom 96 well plates. Recombinant dimeric SIRP α -mouse Fc (Abcam ab221342) was added in duplicate half log 10-point DRC within the range of 1.5 ng/mL to 30 $\mu\text{g/mL}$, and 10 $\mu\text{g/ml}$ of Alex Fluor 647 labeled TPP-23 anti CD47 was used to measure relative CD47 expression. After 30 minutes incubation with recombinant SIRP α -mouse Fc, cells were washed twice with FACS Buffer (BD 554657) and fixed for 10 minutes with 1% paraformaldehyde (EMS 30525-89-4). The fixed and washed cells were then incubated for 30 minutes with Goat anti Mouse Fc Alex Fluor 647 (Jackson ImmunoResearch 115-606-071). After washing twice with FACS Buffer the cells were analyzed on BD FACS Canto. FlowJo V10 analysis software was used to determine Alexa Fluor 647 geometric mean fluorescence intensity (gMFI) for all samples.

References

1. Lebedev, A.A. & Isupov, M.N. Space-group and origin ambiguity in macromolecular structures with pseudo-symmetry and its treatment with the program Zanuda. *Acta Crystallogr D Biol Crystallogr* **70**, 2430-43 (2014).
2. Rebres, R.A., Vaz, L.E., Green, J.M. & Brown, E.J. Normal ligand binding and signaling by CD47 (integrin-associated protein) requires a long range disulfide bond between the extracellular and membrane-spanning domains. *J Biol Chem* **276**, 34607-16 (2001).
3. Emsley, P., Lohkamp, B., Scott, W.G. & Cowtan, K. Features and development of Coot. *Acta Crystallogr D Biol Crystallogr* **66**, 486-501 (2010).

REVIEWERS' COMMENTS

Reviewer #1 (Remarks to the Author):

The authors have answered my questions satisfactorily. I recommend publication.

Reviewer #2 (Remarks to the Author):

This reviewer commends the authors for their revisions of the original manuscript, particularly by adding insightful HDX-MS data and new cell binding experiments with CD47 mutants.

Minor points:

-The biological relevance of the two conformations of CD47 identified by MDS has still not been experimentally tested and remains largely un-discussed. In the absence of any experimental data for the importance of such flexibility to the function of the receptor, the last part of the following sentence should be removed/modified (lines 444-446): It does however provide insights as to how the TM helices are oriented, how the assembly ECD-ECLR is coupled and the basic motions they must undergo for functionality.

-Angle values should be rounded to the nearest unit throughout the manuscript. e.g. 17.5° (line 361), ~62.5° (line 362) and ~22.5° (line 382), and also in Fig. 7.

Reviewer #3 (Remarks to the Author):

In the revised version of their manuscript presenting the first atomic-level structure of the CD47 receptor, the authors have sought to address questions about the functional significance of their structural observations by performing additional experiments. First, the authors use HDX-MS experiments to compare the relative flexibilities of CD47 mutants and WT, providing support for certain observations derived from their crystal structure. For example, they find that helix 3, which packs tightly against other residues in the TM bundle, exhibits very low exchange compared to the rest of the structure. As predicted, Y184A (and to a lesser extent, Y184F) substantially increased the

exposure and accessibility of peptides containing the SWF loop, in agreement with the simulation observation that Y184 controls motion of the ECD.

The authors have crafted a clear story and have addressed my previous comments, and the structure of the CD47 remains of importance to the field.

Major comments:

1. Although the HDX-MS data nicely complements the structure and simulation analysis, these data are still somewhat limited in what they tell us about how CD47 functions (e.g. what is the effect of binding of either SIRP-alpha or known antibodies on the conformation of CD47; how is signal transmitted across the membrane, if at all; etc). For example, I wondered whether the authors could use their HDX-MS system to easily establish whether known binders (e.g. antibodies) have allosteric effects on the transmembrane bundle that are transmitted via the ECD. Nevertheless, it is likely beyond the scope of this paper to ask the authors to carry out additional and extensive functional studies of their system; the structural information already presented in the manuscript is likely worth publishing on its own.

2. Along these lines, for simulations, I previously commented that I wasn't sure whether states 1 and 2 represented 'meta-stable' states on the landscape. Given the authors' EM data (shared in their response), as well as the individual simulation replicates, shown in the supplemental figure, I am fine with the authors' choice to refer to these as macrostates or simply 'states'. Perhaps I should rephrase my original question: why do the authors think that the ECD spontaneously moves away from its initial conformation consistently across all three simulations? The fact that it happens so consistently suggests some sort of strain or instability exists in the S1 state. I wondered whether removing the co-crystallized antibody might represent one such perturbation that acts allosterically to destabilize the orientation of Y184, perhaps by acting on the networks described on p. 17 of the current manuscript. It would be helpful to see a picture, similar to what is shown in Fig. S1D, of the residue-level contacts between the ECD and B6H12 to understand how extensive this contact surface is. Rather than simulating with the antibody, the authors could also consider performing simulations of CD47 with B6H12-interacting residues restrained to their crystallographic positions to assess the effect of the bound antibody on the SWF loop conformation and on Y184. (I do understand that additional simulations take time and computing resources that may not be available.) Note that the single-particle EM data does not rule out the possibility that the B6H12 antibody stabilizes the S1 state. In fact, such a mechanism could be functionally quite interesting.

Minor:

For HDX-MS experiments, (how) was back exchange assessed across the three independent experiments? Please state whether or not a fully deuterated control was included to assess uptake. Also note: line 616-617 that the number of exchangeable amide hydrogens is the number of residues excluding the two N-terminal residues *and* excluding proline residues.

Please also include whether or not N-acetyl glucosamines (NAG) modifications were included in simulation.

P. 12-13: I found myself a bit confused by the comparison of human CD47 sequences to different sets of closely related virus analogues (line. 283-284) vs. other viruses found on the phylogenetic tree (l. 276-277). Could the authors better articulate what is meant by these differences?

Fig. 5: please label residue numbers on the structures in A/B or region identifiers (e.g. 'Helix III') in C/D to demonstrate the correspondence between structure and peptides observed by mass spec.

Fig. 6: please remove the 'backwards' arrow or make it dotted, so as to not imply that you observed transitions from S2 back to S1 in simulation.

Reviewer #4 (Remarks to the Author):

In this article, the authors obtain the crystal structure of the full-length receptor CD47. To complement their structural studies, they perform MD simulations and differential HDX measurements.

The HDX-MS data is convincing. It does not strictly follow the recommendations provided in a recent "white paper" for HDX-MS measurements (<https://www.nature.com/articles/s41592-019-0459-y>) (no biological replicates for example). But since the use of the HDX data in the context of this research is mostly qualitative and in support of the structural data, I don't find this to be an issue.

Could the authors specify the sequence coverage they obtained and what filtering parameters they used in PLGS? Looking at the raw HDX data, it appears that there's no peptide redundancy for the region 184-202 and since the filtering parameters are not specified it's hard to tell how likely it is that these peptides are false positives. I would suggest not commenting on these peptides and remove peptide 184-195 from fig.5 since the data supporting the observation is not solid enough. On

the other hand, I find the differential HDX plots showing the effect of C245S and Y184 mutations very informative (currently in sup. Fig.7) and I suggest putting those in the main figure instead of the uptake plots of row c.

I would also refrain from using statements like "...and resulted in a specific increase in the conformational dynamics" (line 317 p14) and replace it by something along the lines "The increase in HD exchange is indicative/suggests an increase in conformational dynamics". It might sound like nit-picking but it is important to bear in mind that HD exchange reflects the stability of the H-bond and therefore an increase in HD can be caused by other things that increased conformational dynamics.

Another minor comment is that presenting the data in supplementary table at the end of the article is a bit unpractical. Why not export the uptake plots directly from DynamX? There's an option to do just that.

Chloe Martens

REVIEWERS' COMMENTS

Reviewer #2 (Remarks to the Author):

This reviewer commends the authors for their revisions of the original manuscript, particularly by adding insightful HDX-MS data and new cell binding experiments with CD47 mutants.

Minor points:

-The biological relevance of the two conformations of CD47 identified by MDS has still not been experimentally tested and remains largely un-discussed. In the absence of any experimental data for the importance of such flexibility to the function of the receptor, the last part of the following sentence should be removed/modified (lines 444-446): It does however provide insights as to how the TM helices are oriented, how the assembly ECD-ECLR is coupled and the basic motions they must undergo for functionality.

We thank the reviewer for pointing this out. Based on our current understanding of the structure and dynamics of the full length CD47, we agree that it would be appropriate to delete the last part of the above-mentioned sentence. The sentence now reads ... *"It does however provide insights as to how the TM helices are oriented and how the assembly ECD-ECLR is coupled."* ...

-Angle values should be rounded to the nearest unit throughout the manuscript. e.g. 17.5° (line 361), ~62.5° (line 362) and ~22.5° (line 382), and also in Fig. 7.

We have rounded the angle values to the nearest unit throughout the manuscript including Fig. 7.

Reviewer #3 (Remarks to the Author):

In the revised version of their manuscript presenting the first atomic-level structure of the CD47 receptor, the authors have sought to address questions about the functional significance of their structural observations by performing additional experiments. First, the authors use HDX-MS experiments to compare the relative flexibilities of CD47 mutants and WT, providing support for certain observations derived from their crystal structure. For example, they find that helix 3, which packs tightly against other residues in the TM bundle, exhibits very low exchange compared to the rest of the structure. As predicted, Y184A (and to a lesser extent, Y184F) substantially increased the exposure and accessibility of peptides containing the SWF loop, in agreement with the simulation observation that Y184 controls motion of the ECD.

The authors have crafted a clear story and have addressed my previous comments, and the structure of the CD47 remains of importance to the field.

Major comments:

1. Although the HDX-MS data nicely complements the structure and simulation analysis, these data are still somewhat limited in what they tell us about how CD47 functions (e.g. what is the effect of binding of either SIRP-alpha or known antibodies on the conformation of CD47; how is signal transmitted across the membrane, if at all; etc). For example, I wondered whether the authors could use their HDX-MS system to easily establish whether known binders (e.g. antibodies) have allosteric effects on the transmembrane bundle that are transmitted via the ECD. Nevertheless, it is likely beyond the scope of this paper to ask the authors to carry out additional and extensive functional studies of their system; the structural information already presented in the manuscript is likely worth publishing on its own.

We thank the reviewer for suggesting experiments to investigate if binding of known protein partners have allosteric effects on the TMD that are transmitted via the ECD. Exploring the potential allosteric effects of binders will be an important step to develop a deeper understanding of CD47 signaling, but such experiments are beyond the scope of this paper.

2. Along these lines, for simulations, I previously commented that I wasn't sure whether states 1 and 2 represented 'meta-stable' states on the landscape. Given the authors' EM data (shared in their response), as well as the individual simulation replicates, shown in the supplemental figure, I am fine with the authors' choice to refer to these as macrostates or simply 'states'. Perhaps I should rephrase my original question: why do the authors think that the ECD spontaneously moves away from its initial conformation consistently across all three simulations? The fact that it happens so consistently suggests some sort of strain or instability exists in the S1 state. I wondered whether removing the co-crystallized antibody might represent one such perturbation that acts allosterically to destabilize the orientation of Y184, perhaps by acting on the networks described on p. 17 of the current manuscript. It would be helpful to see a picture, similar to what is shown in Fig. S1D, of the residue-level contacts between the ECD and B6H12 to understand how extensive this contact surface is. Rather than simulating with the antibody, the authors could also consider performing simulations of CD47 with B6H12-interacting residues restrained to their crystallographic positions to assess the effect of the bound antibody on the SWF loop conformation and on Y184. (I do understand that additional simulations take time and computing resources that may not be available.) Note that the single-particle EM data does not rule out the possibility that the B6H12 antibody stabilizes the S1 state. In fact, such a mechanism could be functionally quite interesting.

We thank the reviewer for the thoughtful comments and suggestions. We agree that it is not possible to completely rule out the possibility that B6H12 can allosterically stabilize the S1 state observed in both the single particle Cryo-EM data and in the crystal structure. It does however rule out any crystallographic effect mediated by B6H12 that may stabilize the S1 state within the crystal lattice. It would be very interesting to uncover a potential mechanism of allosteric stabilization of S1. However, it would be necessary extensive experiments (in addition to MD simulations) to correctly delineate CD47 mechanisms, and this is beyond the scope of this manuscript.

Minor:

For HDX-MS experiments, (how) was back exchange assessed across the three independent experiments? Please state whether or not a fully deuterated control was included to assess uptake. Also note: line 616-617 that the number of exchangeable amide hydrogens is the number of residues excluding the two N-terminal residues *and* excluding proline residues.

We did not perform fully deuteration as the control for back exchange calculation. The % deuterium uptake was calculated based on the theoretical number of exchangeable amide hydrogens of each peptide. Since the HDX comparisons were done qualitatively, the variation of back exchange of each peptide will have negligible impact on the overall conclusion. We thank the reviewer for mentioning the proline. We did consider the proline residues in the Max. D uptake calculation. This description has now been included in the Methods section of the revised manuscript.

Please also include whether or not N-acetyl glucosamines (NAG) modifications were included in simulation.

N-acetylglucosamines were included in the MD simulations as modeled in the crystal structure. We have included a sentence in the Methods section to clarify that NAG's were included in the MD simulations. The sentence now reads ... *"CD47 model was prepared using the Protein Preparation Wizard module in Maestro (Schrodinger, LLC version 2020.1), and included the N-acetylglucosamines as modeled in the CD47^{BRIL}-B6H12 crystal structure."* ...

P. 12-13: I found myself a bit confused by the comparison of human CD47 sequences to different sets of closely related virus analogues (line. 283-284) vs. other viruses found on the phylogenetic tree (l. 276-277). Could the authors better articulate what is meant by these differences?

We thank the reviewer for the comments. We assembled a list of all available sequences for CD47-like receptors from viruses and aligned them with human CD47. The alignment shows only a few residues with strict sequence identity to the human species (Supplementary Data 3), although numerous consensus motifs were identified (SWF included). We then performed a phylogenetic analysis of all these CD47-like receptors from viruses and human CD47 and found two clades of CD47-like receptors from viruses in the phylogenetic tree (Supplementary Fig. 6b), one of which is closer to the human CD47 in the evolutionary scale. A new pairwise sequence alignment of the closely related viral sequences and human CD47 was then performed. Strict amino acid identity was then clearly observed in many key regions of the receptor (e.g. ECLR, SWF loop, TMD core and in the HC1) as shown in Supplementary Data 4 on the revised manuscript. This suggests potential important roles for these residues in protein folding or perhaps function, since even evolutionarily distant viral receptor orthologues conserved these regions to preserve a fundamental receptor function (immune system evasion). We updated the alignment between all viral CD47-like sequences closely related to human CD47 in the Supplementary Data 4, and the higher sequence identity is very clear.

Fig. 5: please label residue numbers on the structures in A/B or region identifiers (e.g. 'Helix III') in C/D to demonstrate the correspondence between structure and peptides observed by mass spec.

We thank the reviewer for this suggestion. We have now annotated residue numbers and region identifiers so one can easily correlate to peptides presented in the HDX-MS data.

Fig. 6: please remove the 'backwards' arrow or make it dotted, so as to not imply that you observed transitions from S2 back to S1 in simulation.

We replaced the back arrow on Fig. 6 by a dotted black arrow.

Reviewer #4 (Remarks to the Author):

In this article, the authors obtain the crystal structure of the full-length receptor CD47. To complement their structural studies, they perform MD simulations and differential HDX measurements. The HDX-MS data is convincing. It does not strictly follow the recommendations provided in a recent "white paper" for HDX-MS measurements (<https://www.nature.com/articles/s41592-019-0459-y>) (no biological replicates for example). But since the use of the HDX data in the context of this research is mostly qualitative and in support of the structural data, I don't find this to be an issue. Could the authors specify the sequence coverage they obtained and what filtering parameters they used in PLGS? Looking at the raw HDX data, it appears that there's no peptide redundancy for the region 184-202 and since the filtering parameters are not specified it's hard to tell how likely it is that these peptides are false positives. I would suggest not commenting on these peptides and remove peptide 184-195 from fig.5 since the data supporting the observation is not solid enough.

We thank the reviewer for this comment which helps to improve the HDX-MS section. We obtained 68% sequence coverage for the protein constructs, which was originally mentioned in the Supplementary Table 2 legend, and now better described in the replacement Supplementary Table 2 included in the revised version of the manuscript. The PLGS searching parameters have now been included in the Methods section of the manuscript for clarification as suggested by this reviewer. The peptide 184-194 was identified from multiple runs of different batches of the WT CD47^{BRIL} protein construct (and in the experiments performed with the panel of CD47^{BRIL} mutants) and based on the quality of its MS/MS spectrum (shown below), we believe this is a true positive ID.

In addition, we also identified peptide 195-202 (MS/MS spectrum shown below), which is adjacent to peptide 184-194, from multiple runs of different batches of the WT CD47^{BRIL} protein construct and peptide 195-202 has likewise low deuterium uptake. Together with peptide 184-194 they cover almost the entire helix III, in line with the crystallographic observation that this region of helix III is constrained in the middle of the TMD bundle.

On the other hand, I find the differential HDX plots showing the effect of C245S and Y184 mutations very informative (currently in sup. Fig.7) and I suggest putting those in the main figure instead of the uptake plots of row c.

We thank the reviewer for this suggestion. We also agree that the single point HDX plots in Supplementary Fig. 7 are very informative and easy to read. However, we feel that the uptake plots in Fig. 5 are concise, provide better visualization from the kinetic point of view and are more aligned with the overall message of Fig. 5 in the main text. Therefore, we would prefer to keep the differential plots as Supplemental Fig. 7.

I would also refrain from using statements like "...and resulted in a specific increase in the conformational dynamics" (line 317 p14) and replace it by something along the lines "The increase in HD

exchange is indicative/suggests an increase in conformational dynamics". It might sound like nit-picking but it is important to bear in mind that HD exchange reflects the stability of the H-bond and therefore an increase in HD can be caused by other things that increased conformational dynamics.

This is an important point raised by the reviewer and we have reworded that sentence accordingly. We replaced the original sentence ... *"Mutation of the ECL2 residue C245 into a serine prevents formation of the inter-domain disulfide bond (C15-C245), impairs receptor function²⁸ and resulted in a specific increase in the conformational dynamics of the inter-domain ¹¹⁴RVVSWF¹¹⁹ linker (peptides 111-126, 112-126, 114-126 and 119-126), with negligible HDX changes on other characterized regions of the receptor"*... by a revised version which reads ... *"Mutation of the ECL2 residue C245 into a serine prevents formation of the inter-domain disulfide bond (C15-C245), impairs receptor function²⁸ and resulted in an increase in the HDX exchange indicative of an increase in the conformational dynamics of the inter-domain ¹¹⁴RVVSWF¹¹⁹ linker (peptides 111-126, 112-126, 114-126 and 119-126), with negligible HDX changes on other characterized regions of the receptor"*

Another minor comment is that presenting the data in supplementary table at the end of the article is a bit unpractical. Why not export the uptake plots directly from DynamX? There's an option to do just that.

We thank the reviewer for this suggestion. We feel that the Supplementary table 2 in the previous version of the manuscript (Supplementary Data 5 in the revised manuscript) provides a more detailed view of the D uptake of each peptide at each timepoint, which makes the data more accessible to the readers and provides a more comprehensive view of the data.